physiology, biochemistry, ecology

cardiac function, hypoxia, thermal tolerance, temperature, haematology, climate change

**Authors for correspondence:**
Robine H. J. Leeuwis
e-mail: rhjleeuwis@mun.ca
A. Kurt Gamperl
e-mail: kgamperl@mun.ca

# Research on sablefish (*Anoplopoma fimbria*) suggests that limited capacity to increase heart function leaves hypoxic fish susceptible to heat waves

Robine H. J. Leeuwis, Fábio S. Zanuzzo, Ellen F. C. Peroni
and A. Kurt Gamperl

Department of Ocean Sciences, Memorial University of Newfoundland, St John's, NL, Canada A1C 5S7

(iD) RHJL, 0000-0002-6687-4304; FSZ, 0000-0001-9854-8696; AKG, 0000-0002-9852-4819

Studies of heart function and metabolism have been used to predict the impact of global warming on fish survival and distribution, and their susceptibility to acute and chronic temperature increases. Yet, despite the fact that hypoxia and high temperatures often co-occur, only one study has examined the effects of hypoxia on fish thermal tolerance, and the consequences of hypoxia for fish cardiac responses to acute warming have not been investigated. We report that sablefish (*Anoplopoma fimbria*) did not increase heart rate or cardiac output when warmed while hypoxic, and that this response was associated with reductions in maximum $O_2$ consumption and thermal tolerance ($CT_{max}$) of 66% and approximately 3°C, respectively. Further, acclimation to hypoxia for four to six months did not substantially alter the sablefish's temperature-dependent physiological responses or improve its $CT_{max}$. These results provide novel, and compelling, evidence that hypoxia can impair the cardiac and metabolic response to increased temperatures in fish, and suggest that some coastal species may be more vulnerable to climate change-related heat waves than previously thought. Further, they support research showing that cross-tolerance and physiological plasticity in fish following hypoxia acclimation are limited.

## 1. Introduction

More frequent and extreme warming events (i.e. heat waves) are a consequence of climate change [1–3]. The ability of fish to survive exposure to high temperatures depends to a large extent on the capacity of the heart to deliver more $O_2$ to the tissues to meet elevated metabolic demands (i.e. oxygen consumption, $\dot{M}O_2$) [4,5]. For example, heart rate ($f_H$) and cardiac output ($\dot{Q}$; the amount of blood pumped per minute) increase with temperature, and cardiac collapse [6–10] and neural impairment [11] appear to be key factors in determining the upper thermal limit of fishes. The occurrence and severity of hypoxia are also increasing with climate change [1,12], and hypoxia often coincides with high temperatures in coastal environments [12,13], including at aquaculture cage-sites [14–16]. This could severely limit fish thermal tolerance [17]. This is because low $O_2$ levels result in a regulated decrease in $f_H$ (a response known as bradycardia [18]) that, theoretically, may prevent necessary increases in cardiac function ($f_H$ and $\dot{Q}$) when fish are exposed to high temperatures. The effect of hypoxia on thermal tolerance has only been examined in two fish species [17], and this condition's impact on temperature-dependent cardiorespiratory responses is unknown. Further, despite evidence that acclimation (long-term exposure) to hypoxia modifies the cardiorespiratory response to warming at high $O_2$ levels (i.e. under normoxic conditions) considerably [19], there is conflicting information about whether hypoxic acclimation improves thermal

tolerance, and vice versa (i.e. that there is 'cross-tolerance' between these two oxygen-limiting conditions [19–24]). Specifically, the hypothesis that hypoxia acclimation increases upper thermal tolerance when $O_2$ levels are low has not been experimentally tested. These are important questions given the need to understand how physiological plasticity is related to a fish's tolerance to environmental change, and have implications for fish survival and potential shifts in species' distribution with global warming.

In this study, we conducted two experiments on sablefish (*Anoplopoma fimbria*) with the goal of providing insights into the potential consequences of hypoxia on the cardiorespiratory response and susceptibility of fish to acute warming events. The sablefish is a widely distributed species across the North Pacific, that lives in coastal waters as juveniles but successfully inhabits oxygen minimum zones (OMZs; areas with $O_2$ levels of less than $2 \, \mathrm{mg} \, l^{-1}$) as adults [25,26]. Given that this species encounters, and tolerates [27], a wide range of temperatures and $O_2$ levels, it is an ideal model in which to test the above questions/hypotheses with respect to the relationship between acute hypoxia, adaptation to hypoxia and thermal tolerance. In the first experiment, we acclimated sablefish to either 100% (normoxia) or 40% air saturation (hypoxia) at 10°C for four to six months. This is a moderate hypoxic level for the sablefish at this temperature, as it is well above the critical and lethal $O_2$ thresholds (15.8% and 5.4% air saturation, respectively) for this species [27], but also low enough to potentially constrain $O_2$ delivery and trigger a plastic response. Then, we performed surgery to allow for *in vivo* cardiovascular measurements and repeated blood sampling (see below), and recovered both acclimation groups under normoxia for 2 days before exposing them to acute hypoxia (a decrease to 40% air saturation over 2 h) followed by warming (at $2°C \, h^{-1}$) until their critical thermal maximum ($CT_{max}$; the temperature at which the fish could no longer maintain equilibrium). Throughout the experiment, we recorded $\dot{Q}$, and analysed blood $O_2$-carrying capacity (haemoglobin and haematocrit), stress hormone and glucose levels, and lactate (an index of anaerobic metabolism) (electronic supplementary material, figure S1). We also measured $\dot{M}O_2$ using intermittent-flow respirometry, and used $\dot{M}O_2/\dot{Q}$ (the amount of $O_2$ consumed per millilitre of blood pumped) to estimate $O_2$ extraction by the tissues based on rearranging the Fick equation. It should be noted that during warming, 40% air saturation likely becomes a more severe hypoxia level for sablefish, given that a fish's critical $O_2$ threshold ($P_{crit}$) increases with temperature in other species [17,28]. Subsequently, we performed a follow-up experiment involving the same cardiorespiratory measurements, but where warming occurred under normoxia, to better understand the responses observed under hypoxia.

## 2. Methods

### (a) Experimental animals and hypoxia acclimation

Juvenile sablefish (1 g) were acquired from NOAA's Manchester Research Station (Port Orchard, WA, USA) and reared for approximately 1 year at the Dr Joe Brown Aquatic Research Building (JBARB) of MUN in tanks supplied with 10°C seawater with an $O_2$ level of approximately 100% air saturation (157 mmHg; $9.3 \, \mathrm{mg} \, l^{-1}$), and a 12 h light : 12 h dark photoperiod. Procedures used to acclimate the sablefish to hypoxia (approx.

40% air saturation; 63 mmHg; $3.7 \, \mathrm{mg} \, l^{-1}$) have been described previously [29] and are detailed in the electronic supplementary material. The hypoxic acclimation period was four to six months, depending on when the fish were used in the experiments. Water $PO_2$ and temperature in the hypoxic and normoxic tanks during this period were $42.4 \pm 0.1\%$ air saturation and $10.2 \pm 0.0°C$, and $106.2 \pm 0.5\%$ air saturation and $10.3 \pm 0.0°C$, respectively. The fish were fed a marine fish diet (Europa, Skretting) at 0.65% of their body weight $day^{-1}$, and on days when the hypoxia-acclimated fish did not consume their full ration, the normoxia-acclimated fish were fed the same amount. This ensured that there were no nutritional or morphometric differences between the two experimental groups. The weight (approx. 1320 g), length, condition factor, cardiac and splenic masses of the normoxia- and hypoxia-acclimated sablefish at the time of the experiments were similar (see electronic supplementary material, table S4).

### (b) Surgery and recovery

Sablefish were initially anaesthetized by immersion in seawater containing $0.1 \, \mathrm{g} \, l^{-1}$ tricaine methanesulfonate (MS-222; AquaLife TMS, Syndel Laboratories Ltd, Nanaimo, BC, Canada) until ventilatory movements ceased. After weight and fork length were recorded, the fish were transferred to a surgery table where their gills were continuously irrigated with chilled (4°C) and oxygenated seawater containing a maintenance dose of MS-222 ($0.05 \, \mathrm{g} \, l^{-1}$). To measure *in vivo* cardiac function, a Transonic® flow probe (2.0–2.5 S) was placed around the ventral aorta (electronic supplementary material, figure S1), following the procedures described for Atlantic cod [30], but with some modifications for sablefish. To allow for repeated blood sampling, a polyethylene cannula (PE 50; Intramedic®, Becton Dickinson and Co., NJ, USA) filled with heparinized 0.9% saline ($100 \, \mathrm{U} \, ml^{-1}$) was subsequently inserted into the afferent branchial artery of the second or third gill arch (electronic supplementary material, figure S1).

Fish were recovered in aerated anaesthetic-free seawater at 12°C. When ventilation commenced, the fish were individually placed into 19.8 l cylindrical respirometers (20.3 cm in diameter × 61.0 cm long) for 42–46 h to recover from surgery under normoxic conditions (electronic supplementary material, figure S1). The respirometers were submersed in a water table supplied with seawater from a temperature- and $PO_2$-controlled reservoir, which initially kept water in the respirometers at 12°C and approximately 100% air saturation. Surgery was always performed on one fish from each acclimation group (the respirometer they were placed into randomized), so that these two fish could be tested in parallel. More details on surgery and recovery are provided in the electronic supplementary material.

### (c) Hypoxic warming challenge

On the experimental day, the flow probe leads were connected to the Transonic® flow meter, after which the fish were left to settle for 2 h before performing any measurements. To investigate the cardiorespiratory response to warming under acute hypoxia, the water $PO_2$ was first gradually decreased over the course of 2 h to 40% air saturation. Subsequently, the temperature was increased at $2°C \, h^{-1}$ until the fish reached its $CT_{max}$. This rate of heating in $CT_{max}$ tests is widely used in the field of fish physiology to assess acute thermal tolerance, and is considered to be ecologically relevant (e.g. [17]). Cardiac function and $\dot{M}O_2$ were measured at the following points during the protocol: (i) under normoxia (100% air saturation) at 12°C; (ii) under hypoxia (40% air saturation) at 12°C; and (iii) at every 2°C increase thereafter. Cardiac function and $\dot{M}O_2$ were always recorded simultaneously and measured twice at each time point. The temperature at which the heartbeat became arrhythmic/irregular

($T_{arrhythmia}$) was also recorded. Blood (approx. 0.7 ml) was sampled from the cannula immediately after the cardiac function and $\dot{M}O_2$ measurements were taken at four points during the experiment: (i) under normoxia at 12°C; (ii) under hypoxia at 12°C; (iii) under hypoxia at 18°C; (iv) under hypoxia at $CT_{max}$. At each sampling, this volume was replaced by an equal volume of 0.9% saline to help maintain the fish's blood volume. The entire protocol took approximately 11 h to complete. Upon reaching their $CT_{max}$, fish were euthanized as quickly as possible inside their respirometer (by injecting MS-222 into their respirometer through a tube while in 'recirculation' mode: final concentration 0.3 g l$^{-1}$) [27]. This was important as the $CT_{max}$ of the paired fish often differed. After removing the fish from their respirometer, correct placement of the flow probe was verified. Then, the heart and spleen were dissected out, rinsed in saline, blotted dry and weighted to determine the relative atrial, ventricular, bulbus and splenic masses [(tissue mass/body mass) × 100].

## (d) Measurements of cardiorespiratory function

Cardiac function was recorded at 20 Hz by interfacing the flow meter with an MP100A-CE data acquisition system and a laptop running AcqKnowledge software (BIOPAC Systems Inc., Goleta, CA, USA). We calculated $f_H$ (in beats min$^{-1}$) manually by determining the time required for 20 systolic peaks in the blood flow recording. This time period was also used to measure $\dot{Q}$ (in ml min$^{-1}$), which was expressed relative to the fish's body mass (in ml min$^{-1}$ kg$^{-1}$). We calculated $V_S$ (in ml beat$^{-1}$ kg$^{-1}$) as $\dot{Q}/f_H$. Sometimes, small peaks were seen in the recordings (e.g. electronic supplementary material, figure S2). These were not counted as systolic peaks, as they resulted from the Transonic® flow probes picking up ventilatory movements. However, it is interesting that ventilation appeared to remain regular after the heart went arrhythmic (see electronic supplementary material, figure S2$d$,$h$), and this is consistent with recent data suggesting that arrhythmias at high temperatures are not caused by nervous dysfunction, but by ionic disturbance at the level of the ventricular myocytes [31]. The Transonic® flow probes were calibrated prior to the experiments (see electronic supplementary material).

We measured $\dot{M}O_2$ (in mg O$_2$ min$^{-1}$ kg$^{-1}$) with the automated intermittent-flow respirometry system described previously [27,32]. The durations of the 'flushing' and 'recirculation' periods were adjusted throughout the protocol, to ensure an $R^2 > 0.90$ for each measurement, and to avoid a decline in $PO_2$ inside the respirometers of more than approximately 5% air saturation. The respirometers were cleaned regularly to prevent background bacterial respiration. This background was considered negligible based on overnight measurements and experiments using respirometers without fish. In these tests, $PO_2$ declined by less than 1% air saturation during the measurement period. We used $\dot{M}O_2$ per unit of $\dot{Q}$ ($\dot{M}O_2/\dot{Q}$; mg O$_2$ l$^{-1}$ blood) as a measure of tissue O$_2$ extraction, by rearranging the Fick equation:

$$\dot{M}O_2 = \dot{Q} \times (C_aO_2 - C_vO_2),$$

where $C_aO_2$ and $C_vO_2$ are the O$_2$ content of the arterial and venous blood, respectively. We are aware that $\dot{M}O_2/\dot{Q}$ is an indirect measure of O$_2$ extraction and has limitations (i.e. the equation does not account for potential O$_2$ uptake through cutaneous respiration, see [33]), and that for a direct assessment $C_aO_2$ and $C_vO_2$ would need to be measured. However, this approach is used by various research groups (see [19,34–36]), and we measured blood haemoglobin content so that changes in blood O$_2$-carrying capacity could be accounted for in our interpretation of the data. Blood O$_2$ content was not estimated from haemoglobin content, given that the combined effects of temperature, pH and CO$_2$ on Hb–O$_2$ affinity and maximum

saturation of sablefish blood are still unknown. For each fish, the routine and maximum $\dot{M}O_2$ were determined as the average of the $\dot{M}O_2$ measurements at 42–46 h post-surgery, and as the highest $\dot{M}O_2$ recorded during exposure to hypoxic warming, respectively. The scope for $\dot{M}O_2$ was calculated as maximum–routine $\dot{M}O_2$. We are aware that aerobic scope is most commonly determined using swimming-flumes and/or chase protocols, but the use of 'temperature-induced' aerobic scope ($AS_T$) is appropriate for this thermal tolerance study and provides equivalent data to these traditional methods (e.g. [27,37,38]). Resting, and maximum values, and values for scope for cardiac parameters ($f_H$, $\dot{Q}$, $V_S$) and $\dot{M}O_2/\dot{Q}$ were determined in the same way as for $\dot{M}O_2$.

## (e) Blood sampling and analyses

The collected blood samples were immediately aliquoted for the analysis of various haematological parameters. Blood was first drawn into microhaematocrit tubes and these were centrifuged at 10 000$g$ for 5 min to determine Hct (%). An aliquot of 50 µl of blood was collected for the measurement of blood Hb concentration. Then, the remaining blood was centrifuged for 1 min at 10 000$g$ in a mini-centrifuge (05-090-128, Fisher Scientific) and 300 µl of plasma was pipetted into a 1.5 ml brown (opaque) Eppendorf tube containing 15 µl of 0.2 M EDTA and 15 µl of 0.15 M glutathione for later measurement of circulating catecholamine levels. The rest of the plasma was divided into 50 µl aliquots for the measurement of cortisol, lactate and glucose. The RBC pellet was used for the analysis of RBC protein content. All samples were immediately frozen in liquid N$_2$ and stored at −80°C. See electronic supplementary material for further details on the blood sampling and analyses.

## (f) Additional experiment

The primary aim of this experiment was to verify that sablefish have the capacity to enhance $f_H$ and $\dot{Q}$ when warmed under normoxia, like other teleosts [9,10]. The sablefish used for this experiment were acquired from Golden Eagle Sable Fish (Vancouver Island, British Columbia, Canada) as young juveniles, and reared for approximately 1.5 years at the JBARB at the same temperature, $PO_2$ and light regime, and with the same diet and ration, as the sablefish used in the initial experiment. The sablefish were also of a comparable age (less than two months older), and had similar body morphometrics (e.g. the same body mass), although length, condition factor, and atrial and ventricular mass differed (see electronic supplementary material, table S4). The surgical procedures in the additional experiment were exactly the same as in the initial experiment, except that fish were not cannulated, given that this research focused on measuring the fish's cardiac response and the presence/absence of blood sampling did not influence thermal tolerance (electronic supplementary material, figure S3). Furthermore, surgical recovery was only for 1 day (22–23 h) instead of 2 days (42–46 h), because it was shown in the initial experiment that 1 day was adequate for the cardiorespiratory variables to stabilize (see electronic supplementary material). Measurements of cardiac function and $\dot{M}O_2$, and all other procedures, were performed as described for the initial experiment. To maintain normoxia ($PO_2$ at approx. 100% air saturation) throughout the thermal challenge, including the 'recirculation' periods of $\dot{M}O_2$ measurements, O$_2$ was bubbled into the water as required using the control system (OXY-REG, Loligo Systems) that was described previously [27]. More details on the additional experiment are provided in the electronic supplementary material.

## (g) Data statistical analyses

All statistical analyses were performed using Rstudio v. 1.2.5033 with R v. 3.6.2 unless mentioned otherwise. Cardiorespiratory

and haematological data from the initial experiment shown in figures 1, 4 and S4 (electronic supplementary material) were analysed using a general linear mixed model (lmer function), which included fish as a random factor, and acute condition (hypoxia and warming), acclimation condition and their interaction, as fixed effects (see electronic supplementary material for further details). Main effects were assessed using ANOVAs (anova function) with type III sums of squares. If the model indicated a significant fixed effect, then differences between the categories among that effect were analysed using post hoc least-squared means comparisons, with the fdr method (Bonferroni-based) for multiplicity p-value adjustment (emmeans and pairs functions). Cardiorespiratory data from the additional experiment shown in figure 1 were analysed in the same way, except that the model did not include a factor for the effect of acclimation. For the cardiorespiratory, thermal tolerance and morphometric variables shown in figure 2 and electronic supplementary material, figure S3 and tables S3–S5, comparisons were made using two-sided Student's t-tests or two-way ANOVAs, and these analyses were done in GraphPad Prism 8. Statistical significance was set at $p < 0.05$ and all data are shown as means ± s.e.m.

## 3. Results

### (a) Cardiorespiratory response to warming when hypoxic

In normoxia-acclimated sablefish, acute hypoxia resulted in a significant decrease in $f_H$ (from $41.3 \pm 2.2$ to $33.4 \pm 1.8$ beats min$^{-1}$, $p < 0.01$), and surprisingly, $f_H$ declined even further during warming (to $25.3 \pm 1.3$ beats min$^{-1}$, $p < 0.0001$) (figure 1a; electronic supplementary material, figure S2 and tables S1 and S2). Stroke volume ($V_S$; the amount of blood pumped per heartbeat) increased slightly with temperature (figure 1e), but this was not sufficient to compensate for the decrease in $f_H$. Thus, $\dot{Q}$ fell during hypoxia (from $25.5 \pm 2.2$ to $21.1 \pm 1.6$ ml min$^{-1}$ kg$^{-1}$, $p < 0.001$) and remained at this level until it declined further at 22°C (i.e. fish approached their CT$_{max}$) (figure 1c); this response resulting in a negative scope for $\dot{Q}$ (figure 2a; electronic supplementary material, table S3). Hypoxia-acclimated sablefish did not experience bradycardia when exposed to acute hypoxia at 12°C (figure 1a; electronic supplementary material, figure S2 and tables S1 and S2), and this likely contributed to their higher aerobic scope (AS$_T$; maximum–routine $\dot{M}O_2$) when warmed ($p < 0.05$; figure 2b; electronic supplementary material, table S3). Nonetheless, $f_H$ and $\dot{Q}$ also failed to increase with temperature in this group (figure 1a,c), and hypoxia acclimation did not change the temperature at which cardiac arrhythmias began ($T_{arrhythmia}$, approx. 21.6°C) or CT$_{max}$ (approx. 22.2°C) ($p = 0.078$ and $p = 0.124$, respectively) (figure 2d).

There was an unusual relationship between $\dot{M}O_2$ and $\dot{Q}$ in sablefish exposed to hypoxic warming (figure 3). Normally, $\dot{M}O_2$ and $\dot{Q}$ are positively correlated in fishes (e.g. [19]). However, we show that as $\dot{M}O_2$ increased, $\dot{Q}$ either declined (in normoxia-acclimated fish) or remained the same (in hypoxia-acclimated fish) (figure 1c,g), and that there was no significant (negative or positive) correlation between $\dot{M}O_2$ and $\dot{Q}$ in the normoxia- and hypoxia-acclimated groups ($p = 0.064$ and $p = 0.884$, respectively; figure 3). To raise $\dot{M}O_2$ during hypoxic warming without increasing $\dot{Q}$, both groups relied solely, and to the same extent, on enhanced $\dot{M}O_2/\dot{Q}$, which increased from approximately 45–55 to 115–135 mg O$_2$ l$^{-1}$ blood pumped ($p < 0.0001$; figure 1i; electronic

supplementary material, table S3). This was partially mediated by increases in blood haemoglobin levels (by 25%; figure 4a,b), which occurred despite cell swelling that resulted in a decrease in the mean cellular haemoglobin concentration (MCHC; figure 4c) and RBC protein levels (electronic supplementary material, figure S4). However, it is clear that the enhancement of $\dot{M}O_2/\dot{Q}$ was primarily due to augmented O$_2$ uptake by the tissues. Further, while hypoxia-acclimated sablefish had significantly lower plasma levels of adrenaline at CT$_{max}$, other stress hormone (cortisol and noradrenaline) and glucose levels increased during warming to a similar degree in both groups (figure 4f–h). The capacity for anaerobic metabolism was considerable in this species (plasma lactate increased from less than 0.2 to approx. 45 mg dl$^{-1}$) and was initiated at 18°C; indicating that the sablefish experienced insufficient O$_2$ delivery well before its CT$_{max}$. However, this parameter was also not affected by hypoxia acclimation (figure 4d).

### (b) Cardiorespiratory response to warming when normoxic

The finding that $f_H$ and cardiac function were severely constrained during warming when the sablefish was hypoxic is unprecedented. Therefore, we performed an additional experiment with normoxia-acclimated sablefish to verify that this species has the capacity to elevate $f_H$ and $\dot{Q}$ when warmed under normoxia, as is typical for other fishes [9,10]. Indeed, sablefish were able to increase $f_H$ and $\dot{Q}$ by twofold (up to $59.1 \pm 3.5$ beats min$^{-1}$ and $39.7 \pm 4.3$ ml beat$^{-1}$ kg$^{-1}$, respectively, $p < 0.0001$) (figures 1 and 2; electronic supplementary material, figure S2 and tables S1 and S3). Further, $\dot{Q}$ and $\dot{M}O_2$ were positively correlated ($p < 0.001$; figure 3), which is consistent with the relationship that is normally observed for fishes (e.g. [19]). These findings suggest that acute hypoxia prevented the sablefish's normal cardiac response to warming, and that this limited their AS$_T$ and thermal tolerance. For example, CT$_{max}$ was approximately 3°C lower in fish exposed to acute hypoxia when compared with those tested under normoxia (figure 2d), and this reduction in CT$_{max}$ was associated with a much lower AS$_T$ (approx. 59 versus 174 mg O$_2$ h$^{-1}$ kg$^{-1}$) (figure 2b); although values for $\dot{M}O_2/\dot{Q}$ were similar (figures 1i,j and 2c; electronic supplementary material, table S3).

## 4. Discussion

Our results (i) provide additional evidence that cardiac function is linked with temperature-induced aerobic scope and thermal tolerance (figure 2), and further emphasize the role of the heart in delivering O$_2$ to the tissues to meet the fish's metabolic demands when exposed to increased temperatures [4–10]; and (ii) show that hypoxia strongly limits the capacity of the cardiorespiratory system to respond to an acute thermal challenge. However, our study also offers other valuable physiological insights, brings up several questions and has major ecological implications.

### (a) Possible reasons why $f_H$ did not increase during hypoxic warming

A key question, which future studies will need to explore, is why hypoxic sablefish were unable to/did not increase $f_H$ when exposed to rising temperatures (figure 1a; electronic

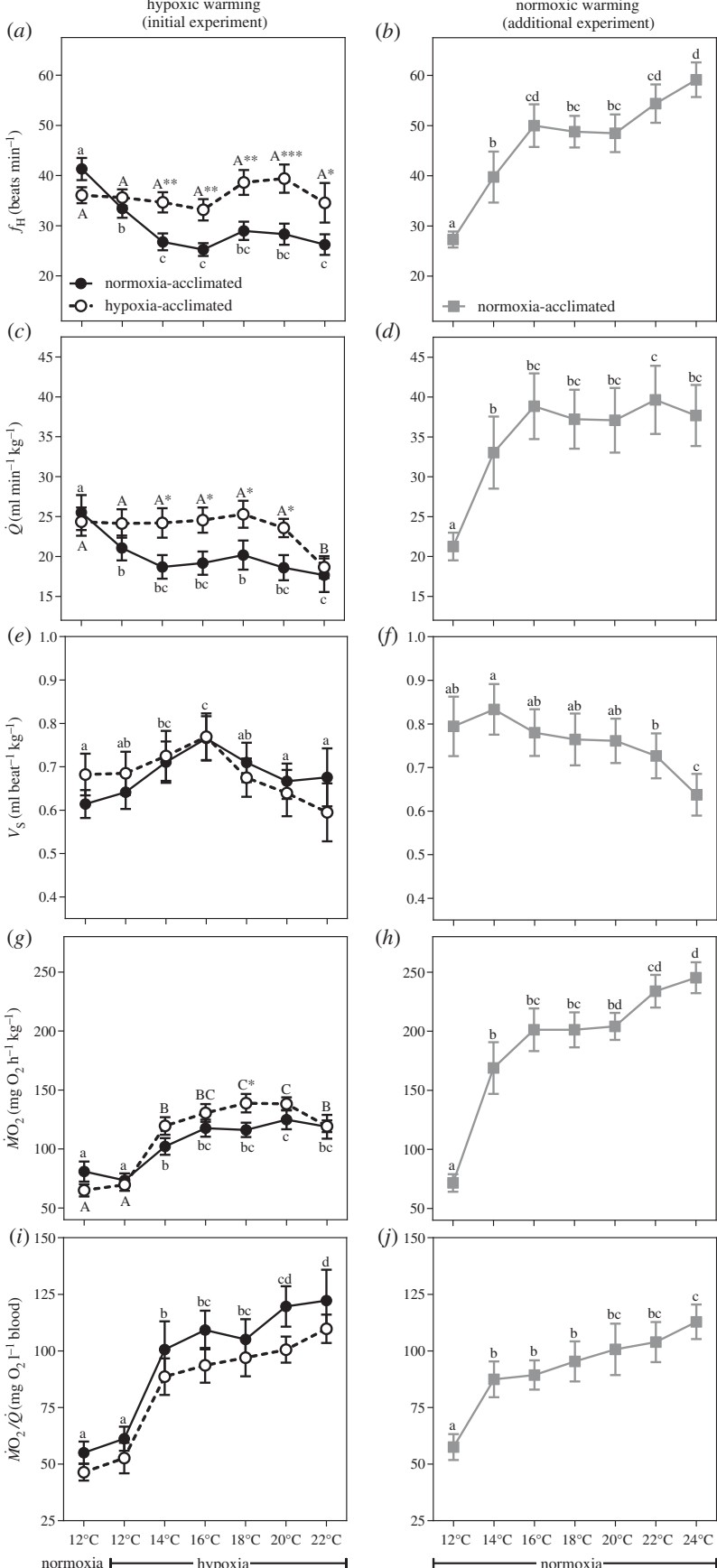

**Figure 1.** Cardiorespiratory responses of normoxia- and hypoxia-acclimated sablefish exposed to hypoxic or normoxic warming. Shown are (a,b) heart rate ($f_H$), (c,d) cardiac output ($\dot{Q}$), (e,f) stroke volume ($V_S$), (g,h) $O_2$ consumption ($\dot{M}O_2$) and (i,j) $O_2$ extraction ($\dot{M}O_2/\dot{Q}$). Hypoxic warming, left panels; normoxic warming, right panels. Symbols without a letter in common are significantly different across the sampling points/temperatures ($p < 0.05$). In the case of a significant acclimation effect and/or acclimation × sampling interaction, lower and uppercase letters indicate differences within the normoxia- and hypoxia-acclimated groups, respectively. Asterisks indicate significant differences between acclimation groups at a particular sampling point (*$p < 0.05$; **$p < 0.01$; ***$p < 0.001$). Values are means ± s.e.m. with $n = 14$–15 per group for the initial experiment (except at 22°C where $n = 9$–10 per group, because 4–6 fish had already reached their $CT_{max}$), and with $n = 9$ for the additional experiment (except at 24°C where $n = 8$, because one fish had already reached its $CT_{max}$).

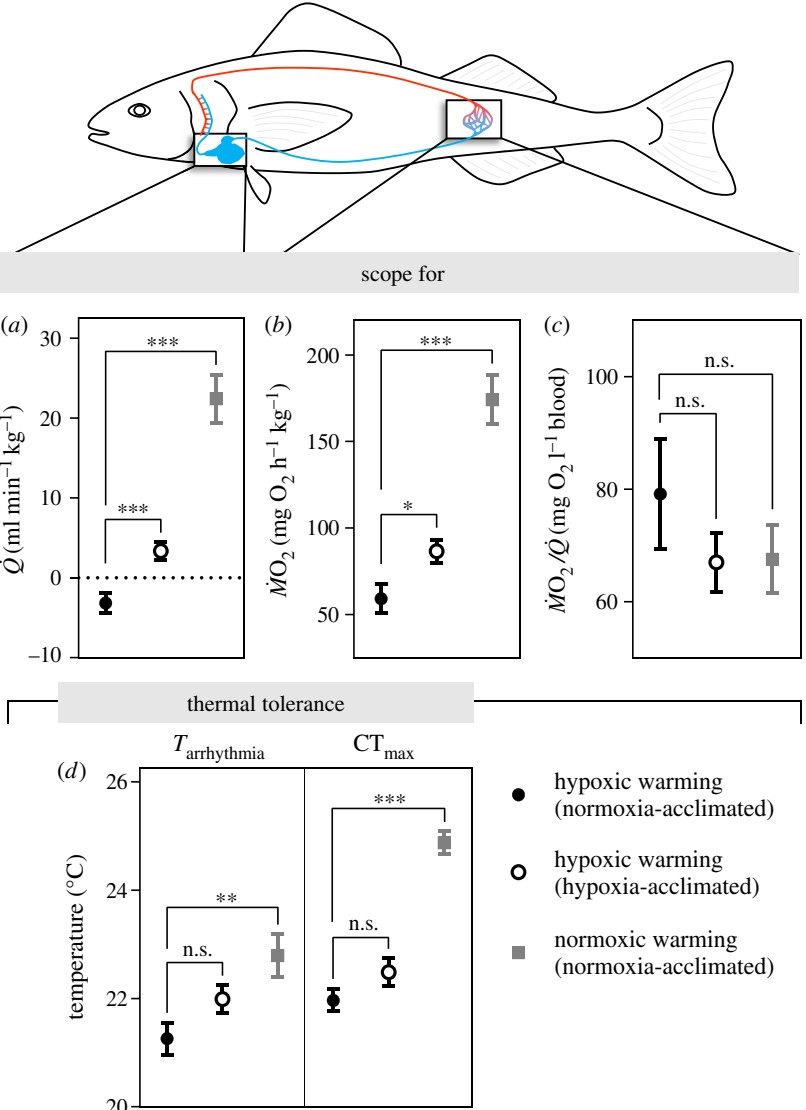

**Figure 2.** The capacity of normoxia- and hypoxia-acclimated sablefish to increase cardiorespiratory function (i.e. heart function and oxygen consumption/extraction by the tissues) when exposed to hypoxic and normoxic warming, and parameters related to thermal tolerance. Values for scope were calculated as maximum—resting values for (a) cardiac output ($\dot{Q}$), (b) $O_2$ consumption ($\dot{M}O_2$) and (c) $O_2$ extraction ($\dot{M}O_2/\dot{Q}$). (d) Parameters for thermal tolerance are the onset temperature of cardiac arrhythmias ($T_{arrhythmia}$) and the critical thermal maximum ($CT_{max}$). The normoxia-acclimated fish tested under hypoxia are compared to the other two groups (n.s. = $p > 0.05$; *$p < 0.05$; **$p < 0.01$; ***$p < 0.001$). Values are means ± s.e.m. with $n = 14$–$15$ and $n = 9$ per group for hypoxic and normoxic warming, respectively. (Online version in colour.)

supplementary material, table S3). One of the potential explanations is that adenosine (which is produced by the heart muscle when $O_2$ supply is limited) depressed $f_H$ [39,40]. However, because hypoxic bradycardia in fish is induced by an increase in cholinergic nervous tone [9,18] on the heart, it is most likely that the sablefish was unable to remove (or reduce) this nervous tone as temperature rose and that this prevented increases in $f_H$. Thus, it appears that while cholinergic inhibition of the heart (i.e. which prevents $f_H$ from getting too high, and delays the onset of arrhythmias) may be beneficial for upper temperature tolerance in species such as rainbow trout (*Oncorhynchus mykiss*) while normoxic [41], it may reduce the thermal tolerance of at least some fishes during hypoxia. This is not only because $f_H$ does not increase in hypoxic sablefish with temperature, but also that $V_S$ increased only slightly or did not change (figure 1e), and this prevented $\dot{Q}$ from increasing with temperature. This finding is in contrast to Keen *et al.* [42] who used the pharmacological agent zatebradine to prevent $f_H$ increases in normoxic rainbow trout during a $CT_{max}$ test,

and showed that increases in $V_S$ completely compensated for the inability to elevate $f_H$. This difference between studies is likely due to the direct negative effects that low $O_2$ levels have on the contractility of the fish heart [43,44].

## (b) New perspectives on fish cardiorespiratory physiology and thermal tolerance

Currently, the literature on fish cardiorespiratory physiology and thermal tolerance suggests that increases in $f_H$ (and thus $\dot{Q}$) are primarily responsible for increases in $\dot{M}O_2$ with temperature and key to a fish's ability to tolerate warming events [4–10], while the importance of enhancing $O_2$ extraction is seldom acknowledged/reported. One reason for this may be that salmonid species are often used as a model for the teleost fish's response to increasing temperatures. For example, the increase in $\dot{M}O_2$ in rainbow trout with temperature is due to a 154% enhancement of $\dot{Q}$ but only a 16% increase in $\dot{M}O_2/\dot{Q}$ (see figs 2 and 4 in Motyka *et al.* [19]). Similarly, Eliason *et al.* [45] reported

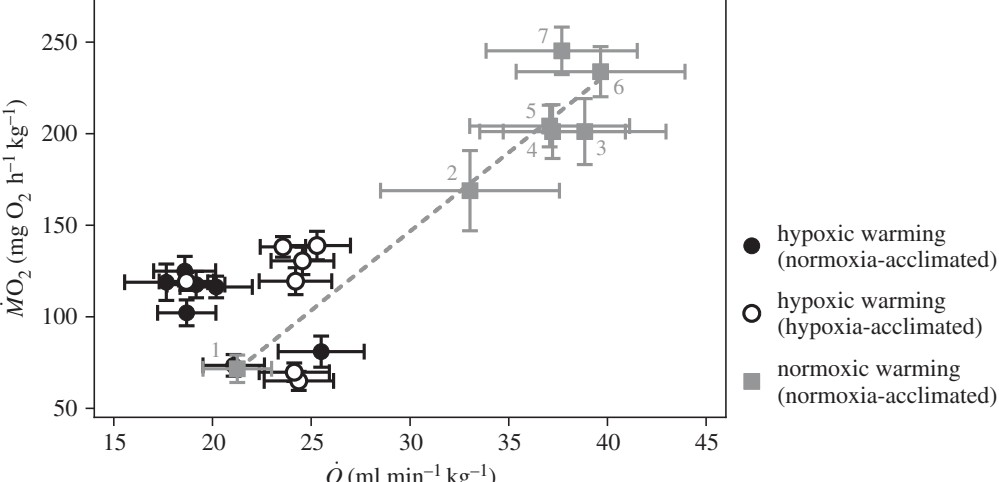

**Figure 3.** The relationship between $\dot{M}O_2$ and $\dot{Q}$ in normoxia- and hypoxia-acclimated sablefish exposed to hypoxic and normoxic warming. No significant linear regression could be fitted to the data for the normoxia and hypoxia acclimation groups tested under hypoxia ($p = 0.064$ and $p = 0.884$, respectively), whereas a significant linear regression was fitted to the data for the fish tested under normoxia ($p < 0.001$, $r^2 = 0.918$, $y = 8.6x - 112.0$). The symbols for the latter group are labelled with grey numbers, which indicate the following conditions: 1, 12°C; 2, 14°C; 3, 16°C; 4, 18°C; 5, 20°C; 6, 22°C; 7, 24°C at normoxia. Values are means ± s.e.m. with $n = 14$–$15$ per group for the hypoxic warming experiment (except at 22°C where $n = 9$–$10$ per group, because 4–6 fish had already reached their $CT_{max}$), and with $n = 9$ for the normoxic warming experiment (except at 24°C where $n = 8$, because one fish had already reached its $CT_{max}$).

that $O_2$ extraction contributed little to the swimming and metabolic performance of sockeye salmon (*Oncorhynchus nerka*) at high temperatures. By contrast, the rise in the sablefish's $\dot{M}O_2$ during normoxic warming was due to comparable increases in $\dot{Q}$ and $\dot{M}O_2/\dot{Q}$ (i.e. both by approx. twofold; figure 1; electronic supplementary material, table S3), and when $\dot{Q}$ did not increase under hypoxia, sablefish relied solely on enhanced $\dot{M}O_2/\dot{Q}$. Therefore, this finding fundamentally changes our understanding of fish cardiorespiratory function and the drivers of $O_2$ consumption at high temperatures. This important physiological adaptation for enhancing $\dot{M}O_2$ in the sablefish (and probably other fishes, e.g. European eel [34]) versus salmonids (trout, char, salmon) may be related to differences in their maximum $f_H$ and scope for increases in $f_H$, and thus $\dot{Q}$. Salmonids acclimated to 10°C, and warmed to their $CT_{max}$, have a maximum $f_H$ and scope for $f_H$ of approx. 120–130 and 70–80 beats min$^{-1}$, respectively [19,46], whereas these values in sablefish are approx. 64 and 37 beats min$^{-1}$ (figure 1a; electronic supplementary material, table S3). What mechanisms allow for the large enhancement in $O_2$ extraction in the sablefish (and possibly other fishes) is not known. However, the sablefish represents one of the five teleostean groups where red blood cells appear to lack the β-adrenergic Na$^+$/H$^+$ exchanger (β-NHE) that protects against intracellular pH reductions [47], and this may allow for enhanced root effect-mediated $O_2$ offloading from haemoglobin at the tissues. Interestingly, some red cell swelling during hypoxic warming was observed in this study (figure 4c; electronic supplementary material, figure S4), although this could have been due to passive/osmotic water influx and increased membrane fluidity/permeability, and is not necessarily indicative of β-NHE [48]. Recently, research has highlighted the potential role of plasma-accessible carbonic anhydrase (PaCA) in tissue $O_2$ extraction in fishes, as this enzyme can acidify the blood as it passes through the tissues, and result in enhanced offloading of $O_2$ from haemoglobin [36]. This mechanism may also play a role in facilitating $O_2$ extraction in sablefish.

## (c) No cross-tolerance and limited plasticity

Motyka *et al.* [19] showed that acclimation of steelhead trout to hypoxia (40% air saturation) does not affect this species' temperature-induced aerobic scope ($AS_T$) or $CT_{max}$ when tested under normoxic conditions, and this largely agrees with the data that we report for the sablefish. Collectively, these data support the majority of research showing that there is no or very little 'cross-tolerance' between these two $O_2$-limiting conditions (i.e. acclimation to hypoxia does not enhance tolerance to high temperatures, and vice versa) [19–21,23,24]. Further, the data from these two experiments fit very well with the 'plastic floors and concrete ceilings' hypothesis [49] which was originally formulated based on the relationship between acclimation temperature and a fish's thermal tolerance. The hypoxia-intolerant rainbow trout in Motyka *et al.* [19] could increase tissue $O_2$ extraction ($\dot{M}O_2/\dot{Q}$) by 74% when acclimated to hypoxia, given that this parameter contributes little to $AS_T$ in normoxia-acclimated fish warmed to their $CT_{max}$. By contrast, the sablefish (which has adapted to the OMZ conditions that they live in as adults [25,26]) has no plasticity to enhance $\dot{M}O_2/\dot{Q}$ further through hypoxic acclimation because it is already a major contributor (i.e. it increased by 117%) to this fish's $AS_T$ (figures 1 and 2; electronic supplementary material, table S3). This suggests that hypoxia-tolerant species may have no, or limited, scope for enhancing their cardiorespiratory capacity upon exposure to oxygen-limiting conditions because it has already been fully exploited as a result of their evolutionary adaptation to environmental hypoxia.

## (d) Concluding remarks and ecological implications of the research

By explicitly testing the effect of an acute warming event on cardiac function when fish are already experiencing moderate hypoxia, we were able to show that prior exposure to acute hypoxia prevents the sablefish from increasing $f_H$ when subsequently exposed to high temperatures (figure 1). These data

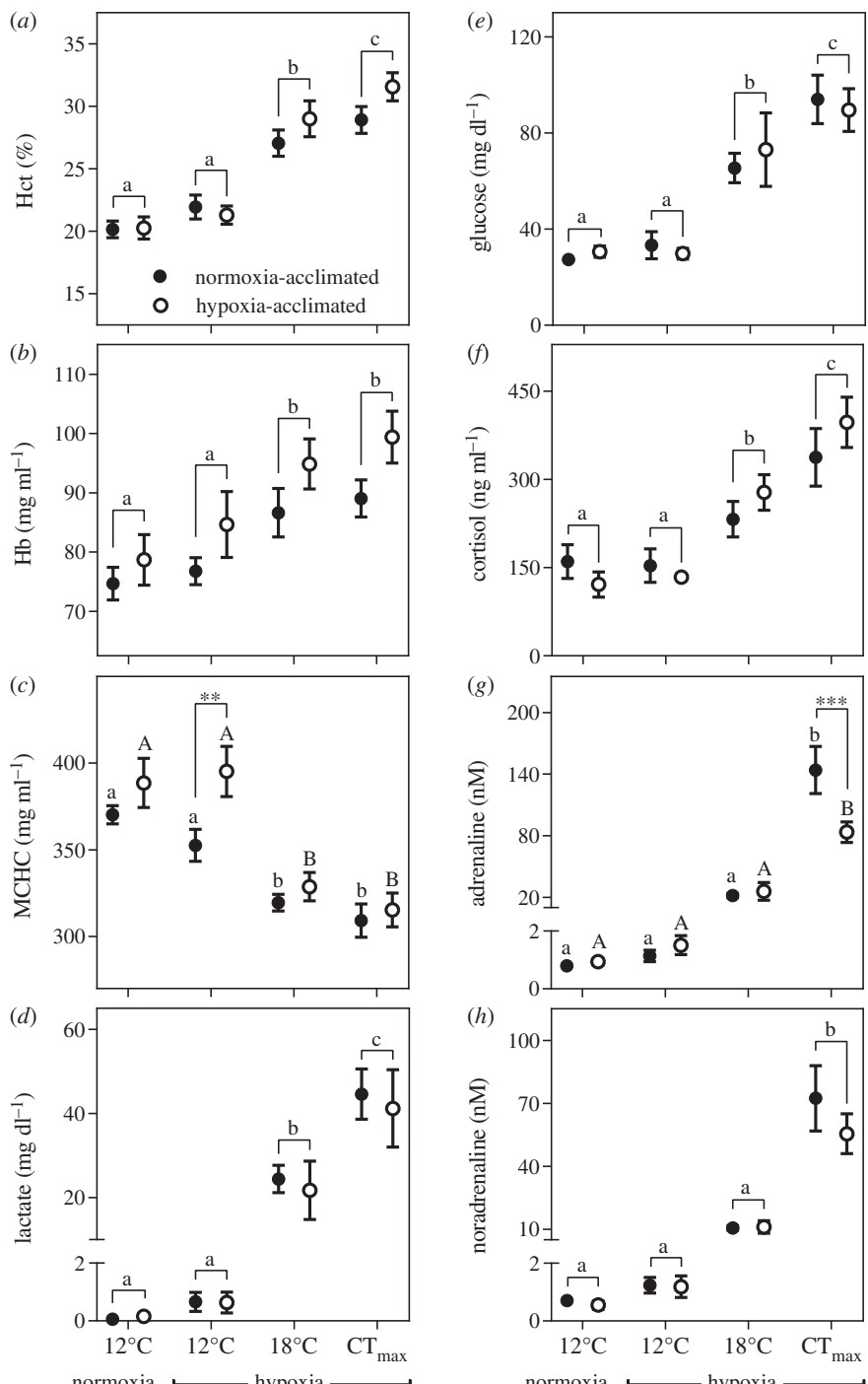

**Figure 4.** Haematological changes in normoxia- and hypoxia-acclimated sablefish when exposed to hypoxic warming. Shown are blood/plasma levels of (*a*) haematocrit (Hct), (*b*) haemoglobin (Hb), (*c*) mean cellular haemoglobin concentration (MCHC), (*d*) lactate, (*e*) glucose, (*f*) cortisol, (*g*) adrenaline and (*h*) noradrenaline. $CT_{max}$, critical thermal maximum. Symbols without a letter in common are significantly different across the sampling points/temperatures ($p < 0.05$). In the case of a significant acclimation effect and/or acclimation × sampling interaction, lower and uppercase letters indicate differences within the normoxia- and hypoxia-acclimated groups, respectively. Asterisks indicate significant differences between acclimation groups at a particular sampling point (**$p < 0.01$; ***$p < 0.001$). Values are means ± s.e.m. with $n = 9$–10 per group.

suggest that the 'physiological chain of command' (in this case nervous control of $f_H$) may be detrimental under certain environmental conditions, or when they are experienced in a particular order (e.g. hypoxia before acute warming). In other words, the depression of $f_H$ by hypoxia appears to dominate over the requirement for increasing $f_H$ during warming, and this severely limits the fish's cardiac response to the latter stressor, and ultimately, appears to constrain thermal tolerance (figure 2); a finding that has recently been confirmed in the Atlantic salmon (*Salmo salar*) (Gamperl *et al.* [50]).

Further, our results indicate that the importance of $O_2$ extraction in determining the thermal tolerance of fishes has been underappreciated, and raise important questions (and testable hypotheses) about: the control of heart rate (function) when high temperatures and hypoxia occur simultaneously; and the role of PaCA in $O_2$ extraction in fishes under varying conditions.

Importantly, these findings also confirm the findings of Ern *et al.* [17] that hypoxia limits fish acute thermal tolerance (figure 2), and draw attention to the possible role of hypoxia

in determining whether fish species/populations survive in the current era of climate change. Coastal areas are experiencing more severe and frequent heat waves [1–3], and periods of hypoxia [1,12], and it is very likely that fish species in some of these areas will suffer a greater loss of biomass than the 5–17% estimated by Lotze *et al*. [51] for higher trophic level organisms in the oceans. This is because many fisheries and marine ecosystem models do not incorporate such periodic events (e.g. [51,52]), and the interactive effects of temperature and hypoxia on organisms in coastal ecosystems are only starting to be addressed [53,54]. Thus, climate change-induced declines in fish populations may be more severe than predicted to date. In addition, there is little evidence that acclimation to hypoxia substantially improves the tolerance of fishes to high temperatures (i.e. there is no cross-tolerance at the whole animal level) [19–21,24]. Although there are instances where the capability for plasticity in fish is considerable (e.g. [55]), it appears that many species of fish (and other ectothermic animals) have limited plasticity to respond to these climate-related challenges (present study, and [19,49,56,57]). Finally, while transgenerational plasticity has been shown to improve some aspects of fish physiology (e.g. aerobic scope) following acclimation to hypoxia or high temperatures, it is unlikely that this will translate into an increased ability to tolerate higher temperatures [56–59]. Thus, it is uncertain whether cross-tolerance, developmental plasticity or adaptation will allow coastal fish populations to persist across their current ranges as marine heat waves and hypoxic zones become more common, and extreme, with climate change.

**Ethics.** All experimental procedures were performed in compliance with the guidelines of the Canadian Council on Animal Care and were approved by the Memorial University of Newfoundland (MUN) Animal Care Committee under the protocol no. 16-92-KG.

**Data accessibility.** All data are provided within the manuscript itself or the electronic supplementary material, and full datasets supporting the main findings of this study are openly available at Zenodo via http://doi.org/10.5281/zenodo.3934340.

**Authors' contributions.** A.K.G. and R.H.J.L. conceptualized the study. R.H.J.L., E.F.C.P. and F.S.Z. performed the experiments and collected the data. A.K.G., R.H.J.L. and F.S.Z. contributed to data analysis and interpretation, and to the writing of the manuscript.

**Competing interests.** Authors declare no competing interests.

**Funding.** This research was supported by an NSERC Discovery Grant to A.K.G. (2016-0448), and research funding to A.K.G. from the Ocean Frontier Institute, through an award from the Canada First Research Excellence Fund. R.H.J.L. was also supported by a Memorial University of Newfoundland, School of Graduate Studies, fellowship.

**Acknowledgements.** We thank Rick Goetz (NOAA, Manchester Research Station, Port Orchard, WA, USA) and Golden Eagle Sablefish for supplying the sablefish for this study, the staff of the Cold-Ocean and Deep-Sea Research Facility and the Dr Joe Brown Aquatic Research Building for help with fish care, and Drs Doug Syme, Tony Farrell and Amanda Bates for providing valuable comments on earlier drafts of this manuscript.

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
