## [Peer Review File · Proceedings of the Royal Society B: Biological Sciences]

Review History

RSPB-2020-1652.R0 (Original submission)

Review form: Reviewer 1

Recommendation

Reject – article is scientifically unsound

Scientific importance: Is the manuscript an original and important contribution to its field?

Marginal

General interest: Is the paper of sufficient general interest?

Good

Quality of the paper: Is the overall quality of the paper suitable?

Marginal

Is the length of the paper justified?

No

Should the paper be seen by a specialist statistical reviewer?

No

Do you have any concerns about statistical analyses in this paper? If so, please specify them explicitly in your report.

No

It is a condition of publication that authors make their supporting data, code and materials available - either as supplementary material or hosted in an external repository. Please rate, if applicable, the supporting data on the following criteria.

Is it accessible?

N/A

Is it clear?

N/A

Is it adequate?

N/A

Do you have any ethical concerns with this paper?

No

Comments to the Author

This study provides interesting data demonstrating that the normal temperature-induced rise in the rate of oxygen consumption, as well as heart rate and cardiac output do not occur when sablefish are exposed to hypoxia. This pattern was similar in fish that had been acclimated to hypoxia for 4-6 months. The study also shows that hypoxia causes a small reduction in the upper critical temperature (CT_{max}) as well as the temperature where heart rate become arrhythmic. The study therefore shows that hypoxia (40% of normal) impairs temperature tolerance, a non-controversial finding with importance for understanding the consequences of current global change scenarios where hypoxia and elevated temperatures often co-occur.

Main objections

1. The authors seem to interpret the findings as if a failure of the cardiovascular system constrains aerobic metabolism, measured as oxygen uptake, but the causality is more likely to be the other way around. Given that metabolism of the hypoxic fish does not increase as much as expected with increased temperature, there are virtually no cardiovascular response (consequently $\dot{M}O_2/Q$ is unaffected by hypoxia). One way to resolve the "hen-egg-problem" of this discussion would have been to measure oxygen concentrations and partial pressures in the arterial venous blood to establish whether hypoxia in fact leads to so low tissue oxygen delivery that metabolism is constrained. This was not done. Instead, the authors estimate arterial venous extraction from the Fick equation (which obviously is perfectly valid if all measurements are correct), but they never relate this calculated extraction to a realistic arterial oxygen concentration that could have been estimated from the Hb concentration and an assumption of oxygen saturation. I'm aware this requires knowledge on the oxygen equilibrium curve of whole blood, but this could have been obtained). The lactate measurement were unfortunately only reported for the highest temperatures and not therefore provide insight to insufficient oxygen delivery.
2. There is no mechanistic explanation for the lack of tachycardia when the fish were heated (see minor comments line 284-287), and it is a great shame that atropine was not injected in some animals to address whether it reflects a hypoxic bradycardia, and hence a regulated response.
3. The title is catchy, but does not adequately address the study. Did you make any perturbation (exercise, stress, drug infusion etc) to verify that cardiac output and heart rate could not be elevated? If not, there is no demonstration of limited capacity; only a demonstration that cardiovascular performance match metabolism.

4. The authors propose that lack of a beta-adrenergic response on the red cells may explain why sablefish seemingly do not offload oxygen. This is unlikely as all other vertebrates – none of which are bestowed with adrenergic stimulation of the Na/H exchanger on their red cells - are quite capable of delivering oxygen in hypoxia. Also, your data on MCHC and red cell protein concentration demonstrates clear and significant signs of swelling indicating a fully functional adrenergic stimulation of the Na/H exchanger and the associated water entry.

5. The level of hypoxia (40%) needs to be clarified. How does this relate to Pcrit at the various temperatures. As the authors are aware, Pcrit increases with temperature and what may be fully sufficient oxygen level at 12C (i.e. above Pcrit at that temperature), may be below Pcrit at 18 and 20C? Again, knowledge on blood oxygen affinity would have been very helpful.

6. The discussion give the impression that a change in dogma is required to understand the influence of temperature on cardiorespiratory physiology and metabolism. I completely disagree. A rise in arterial venous extraction with increased temperature was established by Heath and Hughes (JEB 1973), and has been shown in other species (some of the cited in the submitted ms). Also, given that the submitted ms did NOT measure arterial or venous oxygen, the claim for a new dogma seems excessive to me.

Minor comments (line numbers)

48. A reference to support this claim is needed, also heart function is rather ambiguous (contractility, rate or?).

54. "vital" is rather excessive, consider to tone down the importance of your own contribution?

55-56. something is missing I this sentence? What does "its" refer to?

63. please develop the argument why this species is an ideal model.

66-69. Pcrit depends on temperature so please clarify.

76. two self-citations to a rearranged of the Fick equation are two too many.

80 – methods. The size of the fish at the time of measurements should be included

88. Po₂ does not have the unit of % (it should be kPa, mmHg, torr etc)

97-98. Tricaine methanesulphonate is normally called MS222. Why do you use TMS?

129-134 how many blood samples were taken? The replacement of volume with saline is probably not sufficient to maintain blood volume because saline distributes quickly to the entire extracellular space.

142. describe the setup and delete the self-citation

151. regulation or disturbance?

157. Do not understand "to secure an R₂ of more than 0.90"; the reduction in PO₂ is linear if the rate of oxygen consumption is stable, duration of closure per se has no effect.

274. I don't think you measured aerobic scope, so how can your study address this question?

279. delete "important"

284-287. adenosine is one of many possibilities, so no reason to mention to highlight it. I agree with the authors that vagal tone is a more likely explanation. Why did you not inject atropine or performed cardiac vagotomy?

Figure 3A is not needed as all data reappear in Figure 3B, Also, the difference in the x-axis between the two figures is rather confusing.

Review form: Reviewer 2

Recommendation

Accept with minor revision (please list in comments)

Scientific importance: Is the manuscript an original and important contribution to its field?

Good

General interest: Is the paper of sufficient general interest?

Excellent

Quality of the paper: Is the overall quality of the paper suitable?

Good

Is the length of the paper justified?

Yes

Should the paper be seen by a specialist statistical reviewer?

No

Do you have any concerns about statistical analyses in this paper? If so, please specify them explicitly in your report.

No

It is a condition of publication that authors make their supporting data, code and materials available - either as supplementary material or hosted in an external repository. Please rate, if applicable, the supporting data on the following criteria.

Is it accessible?

Yes

Is it clear?

Yes

Is it adequate?

Yes

Do you have any ethical concerns with this paper?

Yes

Comments to the Author

This is an interesting study that reveals a link between the inability of sablefish to increase heart rate or cardiac output during warming in hypoxic waters and reductions in oxygen consumption rate and thermal tolerance. They also demonstrate the long-term hypoxia acclimation does not substantially impact the relationship between measured variables. A separate experiment is conducted to show the under normoxic conditions, sable fish respond to warming in a similar manner to most fish. Overall, the study is generally well designed and the analytical approaches employed are appropriate. I don't have any substantial concerns with the study, but I would like the authors to address a few points in the manuscript.

Obviously, it would have been preferred if the two studies (hypoxia warming and normoxia warming) were conducted at the same time using fish from the same batch. I understand why the normoxia warming experiment was conducted as an additional experiment, but the authors don't really discuss anywhere in the manuscript the potential issues associated with comparing results from experiments conducted on fish from different sources and done, presumably, at different times. Measurements are made and presented for comparison purposes, but there are differences in various parameters (heart rate) between the groups when held under similar conditions. The authors cannot be certain that the results are directly comparable and that needs to be acknowledged in the manuscript and the implications discussed.

Estimates of MO_{2max} and calculations of aerobic scope. Have the authors verified that the highest MO_2 observed during hypoxia warming is indeed equal to MO_{2max} . Since this is a relatively new species to be investigated in this regard, it would seem a worthwhile endeavour to demonstrate that warm/hypoxic MO_2 s cannot be increased further using other means. Also, on line 173 -I think the authors need to include at warm temperatures in addition to hypoxia for

their estimates of Mo₂max.

62/63 & 66 The units used to describe O₂ for the experiment (% air sat) are different than those used to describe the environmental context of the study (MOX in mg/L). It would be far more convenient for the reader if O₂ was presented in roughly the same units. Even estimating OMZ in % air sat would be helpful.

68. For completeness and readability, it be useful to briefly indicate why surgery was performed.
234. There are differences among the hypoxia/temperature treatments in Figure 1c, so “remained at this level” is not fully accurate.

243 to 246. I think Figure 3a does a poor job of showing the point the authors are trying to make, primarily because no stats are given to support the point. Comparisons between Figure 1C and 1G does a much better job.

274/275. Figure 2 does not explicitly support the “tight” link suggested here. For example, there is no difference in CT_{max} between the two hypoxic warming groups despite differences in cardiac function. Broadly, I would agree there is a link, but the suggestion that it is a tight causal link should be softened.

Decision letter (RSPB-2020-1652.R0)

01-Sep-2020

Dear Miss Leeuwis:

I am writing to inform you that we have now obtained responses from referees on manuscript RSPB-2020-1652 entitled "Limited capacity to increase heart function leaves hypoxic fish susceptible to heat waves" which you submitted to Proceedings B.

Unfortunately, on the advice of the Associate Editor and the referees, your manuscript has been rejected following full peer review. Competition for space in Proceedings B is currently extremely severe, as many more manuscripts are submitted to us than we have space to print. We are therefore only able to publish those that are exceptional, convincing and present significant advances of broad interest, and must reject many good manuscripts.

Please find below the comments received from referees concerning your manuscript, not including confidential reports to the Editor. I hope you may find these useful should you wish to submit your manuscript elsewhere.

We are sorry that your manuscript has had an unfavourable outcome, but would like to thank you for offering your work to Proceedings B.

Sincerely,
Dr Daniel Costa
mailto:proceedingsb@royalsociety.org

Associate Editor
Comments to Author:

Thank you for your patience with the review process. As you can see we have received two expert reviews of your manuscript. Both of these reviewers saw merit in the topic and your approach, and I agree. This is a nicely written manuscript that would be of interest to fish cardiovascular biologists and thermal biologists. Unfortunately, both reviewers provided detailed

comments on their concerns with the work. One reviewer in particular raised several critical issues with the manuscript, and one of these major concerns undermines the conclusions in the paper. Specifically, with the data presented in the manuscript, causality is difficult to discern and the reviewer has requested additional data (measurements of oxygen concentrations and partial pressures in the arterial venous blood) to support or refute your interpretations. I agree that this additional information would significantly improve the quality of the work and the reliability of the conclusions drawn. I am well aware that given the ongoing global crisis, additional experiments are a lot to ask, but without these additions the conclusions drawn from the work are not well grounded in the evidence presented, and the work is thus not a good fit for Proc. B.

Reviewer(s)' Comments to Author:

Referee: 1

Comments to the Author(s)

This study provides interesting data demonstrating that the normal temperature-induced rise in the rate of oxygen consumption, as well as heart rate and cardiac output do not occur when sablefish are exposed to hypoxia. This pattern was similar in fish that had been acclimated to hypoxia for 4-6 months. The study also shows that hypoxia causes a small reduction in the upper critical temperature (CT_{max}) as well as the temperature where heart rate becomes arrhythmic. The study therefore shows that hypoxia (40% of normal) impairs temperature tolerance, a non-controversial finding with importance for understanding the consequences of current global change scenarios where hypoxia and elevated temperatures often co-occur.

Main objections

1. The authors seem to interpret the findings as if a failure of the cardiovascular system constrains aerobic metabolism, measured as oxygen uptake, but the causality is more likely to be the other way around. Given that metabolism of the hypoxic fish does not increase as much as expected with increased temperature, there are virtually no cardiovascular responses (consequently $\dot{M}O_2/Q$ is unaffected by hypoxia). One way to resolve the "hen-egg-problem" of this discussion would have been to measure oxygen concentrations and partial pressures in the arterial venous blood to establish whether hypoxia in fact leads to so low tissue oxygen delivery that metabolism is constrained. This was not done. Instead, the authors estimate arterial venous extraction from the Fick equation (which obviously is perfectly valid if all measurements are correct), but they never relate this calculated extraction to a realistic arterial oxygen concentration that could have been estimated from the Hb concentration and an assumption of oxygen saturation. I'm aware this requires knowledge on the oxygen equilibrium curve of whole blood, but this could have been obtained). The lactate measurements were unfortunately only reported for the highest temperatures and not therefore provide insight to insufficient oxygen delivery.
2. There is no mechanistic explanation for the lack of tachycardia when the fish were heated (see minor comments line 284-287), and it is a great shame that atropine was not injected in some animals to address whether it reflects a hypoxic bradycardia, and hence a regulated response.
3. The title is catchy, but does not adequately address the study. Did you make any perturbation (exercise, stress, drug infusion etc) to verify that cardiac output and heart rate could not be elevated? If not, there is no demonstration of limited capacity; only a demonstration that cardiovascular performance matches metabolism.
4. The authors propose that lack of a beta-adrenergic response on the red cells may explain why sablefish seemingly do not offload oxygen. This is unlikely as all other vertebrates - none of which are bestowed with adrenergic stimulation of the Na/H exchanger on their red cells - are quite capable of delivering oxygen in hypoxia. Also, your data on MCHC and red cell protein concentration demonstrates clear and significant signs of swelling indicating a fully functional adrenergic stimulation of the Na/H exchanger and the associated water entry.
5. The level of hypoxia (40%) needs to be clarified. How does this relate to P_{crit} at the various temperatures. As the authors are aware, P_{crit} increases with temperature and what may be a fully sufficient oxygen level at 12°C (i.e. above P_{crit} at that temperature), may be below P_{crit} at 18 and 20°C? Again, knowledge on blood oxygen affinity would have been very helpful.
6. The discussion gives the impression that a change in dogma is required to understand the influence of temperature on cardiorespiratory physiology and metabolism. I completely disagree.

A rise in arterial venous extraction with increased temperature was established by Heath and Hughes (JEB 1973), and has been shown in other species (some of the cited in the submitted ms). Also, given that the submitted ms did NOT measure arterial or venous oxygen, the claim for a new dogma seems excessive to me.

Minor comments (line numbers)

48. A reference to support this claim is needed, also heart function is rather ambiguous (contractility, rate or?).

54. "vital" is rather excessive, consider to tone down the importance of your own contribution?

55-56. something is missing in this sentence? What does "its" refer to?

63. please develop the argument why this species is an ideal model.

66-69. Pcrit depends on temperature so please clarify.

76. two self-citations to a rearranged of the Fick equation are two too many.

80 - methods. The size of the fish at the time of measurements should be included

88. Po₂ does not have the unit of % (it should be kPa, mmHg, torr etc)

97-98. Tricaine methanesulphonate is normally called MS222. Why do you use TMS?

129-134 how many blood samples were taken? The replacement of volume with saline is probably not sufficient to maintain blood volume because saline distributes quickly to the entire extracellular space.

142. describe the setup and delete the self-citation

151. regulation or disturbance?

157. Do not understand "to secure an R₂ of more than 0.90"; the reduction in PO₂ is linear if the rate of oxygen consumption is stable, duration of closure per se has no effect.

274. I don't think you measured aerobic scope, so how can your study address this question?

279. delete "important"

284-287. adenosine is one of many possibilities, so no reason to mention to highlight it. I agree with the authors that vagal tone is a more likely explanation. Why did you not inject atropine or performed cardiac vagotomy?

Figure 3A is not needed as all data reappear in Figure 3B, Also, the difference in the x-axis between the two figures is rather confusing.

Referee: 2

Comments to the Author(s)

This is an interesting study that reveals a link between the inability of sablefish to increase heart rate or cardiac output during warming in hypoxic waters and reductions in oxygen consumption rate and thermal tolerance. They also demonstrate the long-term hypoxia acclimation does not substantially impact the relationship between measured variables. A separate experiment is conducted to show the under normoxic conditions, sable fish respond to warming in a similar manner to most fish. Overall, the study is generally well designed and the analytical approaches employed are appropriate. I don't have any substantial concerns with the study, but I would like the authors to address a few points in the manuscript.

Obviously, it would have been preferred if the two studies (hypoxia warming and normoxia warming) were conducted at the same time using fish from the same batch. I understand why the normoxia warming experiment was conducted as an additional experiment, but the authors don't really discuss anywhere in the manuscript the potential issues associated with comparing results from experiments conducted on fish from different sources and done, presumably, at different times. Measurements are made and presented for comparison purposes, but there are differences in various parameters (heart rate) between the groups when held under similar conditions. The authors cannot be certain that the results are directly comparable and that needs to be acknowledged in the manuscript and the implications discussed.

Estimates of Mo₂max and calculations of aerobic scope. Have the authors verified that the highest MO₂ observed during hypoxia warming is indeed equal to MO₂max. Since this is a relatively new species to be investigated in this regard, it would seem a worthwhile endeavour to

demonstrate that warm/hypoxic MO₂s cannot be increased further using other means. Also, on line 173 –I think the authors need to include at warm temperatures in addition to hypoxia for their estimates of Mo₂max.

62/63 & 66 The units used to describe O₂ for the experiment (% air sat) are different than those used to describe the environmental context of the study (MO₂ in mg/L). It would be far more convenient for the reader if O₂ was presented in roughly the same units. Even estimating OMZ in % air sat would be helpful.

68. For completeness and readability, it be useful to briefly indicate why surgery was performed.

234. There are differences among the hypoxia/temperature treatments in Figure 1c, so “remained at this level” is not fully accurate.

243 to 246. I think Figure 3a does a poor job of showing the point the authors are trying to make, primarily because no stats are given to support the point. Comparisons between Figure 1C and 1G does a much better job.

274/275. Figure 2 does not explicitly support the “tight” link suggested here. For example, there is no difference in CT_{max} between the two hypoxic warming groups despite differences in cardiac function. Broadly, I would agree there is a link, but the suggestion that it is a tight causal link should be softened.

RSPB-2020-2340.R0

Review form: Reviewer 3 (Christian Damsgaard)

Recommendation

Major revision is needed (please make suggestions in comments)

Scientific importance: Is the manuscript an original and important contribution to its field?

Excellent

General interest: Is the paper of sufficient general interest?

Excellent

Quality of the paper: Is the overall quality of the paper suitable?

Excellent

Is the length of the paper justified?

Yes

Should the paper be seen by a specialist statistical reviewer?

No

Do you have any concerns about statistical analyses in this paper? If so, please specify them explicitly in your report.

No

It is a condition of publication that authors make their supporting data, code and materials available - either as supplementary material or hosted in an external repository. Please rate, if applicable, the supporting data on the following criteria.

Is it accessible?

Yes

Is it clear?

Yes

Is it adequate?

Yes

Do you have any ethical concerns with this paper?

No

Comments to the Author

It was a great pleasure to read the manuscript by Leeuwis and colleagues on the interactions between hypoxia acclimation and high temperature on heart function. Here the authors present an impressive 6-month hypoxia acclimation of sablefish followed by invasive cardiorespiratory measurements in a multistressor study design. This is an impressive feat and rare for ecophysiological studies with global change perspectives. Despite my great enthusiasm for the manuscript, I have a list of queries that may improve the presentation of the manuscript.

Major comments:

1. The introduction reads well, but it does not narrow down to a set of hypotheses that you test in your experiments. You do present one hypothesis that hypoxia acclimation improves $\dot{V}_{O_2}/\dot{V}_{O_2}$ (l 52-54), but this is easy to test and does not require an invasive physiological study. I am sure that you had other specific hypotheses in mind when you designed and executed this sophisticated study, so I urge you to enlighten the reader with your motivation for the study. You actually hint to the fact that there are multiple questions/hypotheses (e.g., line 54), but it is not clear to me what these additional hypotheses are.

2. Scope: You calculate scope, such as aerobic scope, as maximum measured in hypoxia across all temperatures minus the average RMR at 12 degree C in normoxia. You need to justify why this calculation is valid:

2.1 Would it not be more valid to use SMR rather than RMR to calculate scope, which you could extract as a lower percentile of MR over the recovery period in the respirometer?

2.2 MMR was determined as the maximum MR over the experimental design. However, MMR changes with temperature, so you cannot calculate scope when "SMR" and MMR were measured at different temperatures.

2.3 You need to justify your specific protocol can be used to determine MMR without the inclusion of a chase protocol or exhaustive swimming.

Minor comments

l 31-32: you only show this in one species, so I do not see how your data can justify this extrapolation across coastal species.

l. 93-94: Also include that heart and spleen mass did not differ.

$\dot{M}O_2/\dot{Q}$ provides an absolute measure of tissue oxygen extraction, but to get a feeling for how much of the oxygen-carrying capacity (OCC) is extracted at the tissues, you could consider calculate $\dot{M}O_2/\dot{Q}/OCC$. You hint to this on l. 170-171, but I do not see this in the text.

Figure 1 and 4. The scaling of the x-axis is indicated by color and text. Remove one of them - I suggest removing the color.

l. 197: No, the fish in the additional experiment had longer fork length, lower condition factor, and higher heart masses. This should be noted here. You should also mention in the discussion if you expect these differences to affect the interpretations of your results.

What is the purpose of the fish in fig 2 other than showing where the heart is positioned in a fish? I suggest removing it.

Very minor comment

You report oxygen partial pressures in %, mg l^{-1} , and kPa. Be consistent throughout the text, and consider using mmHg as many respiratory physiologists may read the paper. Also, consider using mM/ μM / nM for concentrations of Hb, lactate, MCHC, glucose, NH_3 , and cortisol.

Decision letter (RSPB-2020-2340.R0)

12-Oct-2020

Dear Miss Leeuwis:

Your manuscript has now been peer reviewed by a third referee. This referee was positive but still had some issues that he/she felt needed to be addressed. However, as this third review was based on an appeal of an earlier decision with a positive and very negative review, we are requesting that you revise your manuscript and respond to the comments of all three reviews. The third reviewers' comments (not including confidential comments to the Editor) are included at the end of this email for your reference.

Research ethics:

Use of animals and field studies:

It is a condition of publication that you make available the data and research materials supporting the results in the article (<https://royalsociety.org/journals/authors/author-guidelines/#data>). Datasets should be deposited in an appropriate publicly available repository and details of the associated accession number, link or DOI to the datasets must be included in the Data Accessibility section of the article (<https://royalsociety.org/journals/ethics-policies/data-sharing-mining/>). Reference(s) to datasets should also be included in the reference list of the article with DOIs (where available).

Please submit a copy of your revised paper within three weeks. If we do not hear from you within this time your manuscript will be rejected. If you are unable to meet this deadline please let us know as soon as possible, as we may be able to grant a short extension.

Best wishes,

Dr Daniel Costa

Reviewer(s)' Comments to Author:

Referee: 3

Comments to the Author(s).

It was a great pleasure to read the manuscript by Leeuwis and colleagues on the interactions between hypoxia acclimation and high temperature on heart function. Here the authors present an impressive 6-month hypoxia acclimation of sablefish followed by invasive cardiorespiratory measurements in a multistressor study design. This is an impressive feat and rare for ecophysiological studies with global change perspectives. Despite my great enthusiasm for the manuscript, I have a list of queries that may improve the presentation of the manuscript.

Major comments:

1. The introduction reads well, but it does not narrow down to a set of hypotheses that you test in your experiments. You do present one hypothesis that hypoxia acclimation improves CT_{max} (l 52-54), but this is easy to test and does not require an invasive physiological study. I am sure that you had other specific hypotheses in mind when you designed and executed this sophisticated study, so I urge you to enlighten the reader with your motivation for the study. You actually hint to the fact that there are multiple questions/hypotheses (e.g., line 54), but it is not clear to me what these additional hypotheses are.

2. Scope: You calculate scope, such as aerobic scope, as maximum measured in hypoxia across all temperatures minus the average RMR at 12 degree C in normoxia. You need to justify why this calculation is valid:

2.1 Would it not be more valid to use SMR rather than RMR to calculate scope, which you could extract as a lower percentile of MR over the recovery period in the respirometer?

2.2 MMR was determined as the maximum MR over the experimental design. However, MMR changes with temperature, so you cannot calculate scope when "SMR" and MMR were measured at different temperatures.

2.3 You need to justify your specific protocol can be used to determine MMR without the inclusion of a chase protocol or exhaustive swimming.

Minor comments

l 31-32: you only show this in one species, so I do not see how your data can justify this extrapolation across coastal species.

l. 93-94: Also include that heart and spleen mass did not differ.

MO_{2}/Q provides an absolute measure of tissue oxygen extraction, but to get a feeling for how much of the oxygen-carrying capacity (OCC) is extracted at the tissues, you could consider calculate $MO_{2}/Q/OCC$. You hint to this on l. 170-171, but I do not see this in the text.

Figure 1 and 4. The scaling of the x-axis is indicated by color and text. Remove one of them - I suggest removing the color.

l. 197: No, the fish in the additional experiment had longer fork length, lower condition factor, and higher heart masses. This should be noted here. You should also mention in the discussion if you expect these differences to affect the interpretations of your results.

What is the purpose of the fish in fig 2 other than showing where the heart is positioned in a fish? I suggest removing it.

Very minor comment

You report oxygen partial pressures in %, mg l⁻¹, and kPa. Be consistent throughout the text, and consider using mmHg as many respiratory physiologists may read the paper. Also, consider using mM/μM/nM for concentrations of Hb, lactate, MCHC, glucose, NH₃, and cortisol.

Author's Response to Decision Letter for (RSPB-2020-2340.R0)

See Appendix A.

RSPB-2020-2340.R1 (Revision)

Review form: Reviewer 4

Recommendation

Reject – article is not of sufficient interest (we will consider a transfer to another journal)

Scientific importance: Is the manuscript an original and important contribution to its field?

Good

General interest: Is the paper of sufficient general interest?

Acceptable

Quality of the paper: Is the overall quality of the paper suitable?

Acceptable

Is the length of the paper justified?

Yes

Should the paper be seen by a specialist statistical reviewer?

No

Do you have any concerns about statistical analyses in this paper? If so, please specify them explicitly in your report.

No

It is a condition of publication that authors make their supporting data, code and materials available - either as supplementary material or hosted in an external repository. Please rate, if applicable, the supporting data on the following criteria.

Is it accessible?

Yes

Is it clear?

Yes

Is it adequate?

Yes

Do you have any ethical concerns with this paper?

No

Comments to the Author

Leeuwis and colleagues studied the effects of warming on cardiometabolic performance in sablefish during hypoxic versus normoxic conditions. They saw that during hypoxia heart rate and cardiac output did not change during warming, but during normoxia these variables increased. The authors conclude that the inability to increase heart rate and cardiac output during

hypoxia constrained thermal tolerance.

Major concerns:

The authors bluntly state, e.g. in the 'running head', that 'hypoxic fish CANNOT increase heart rate'. I disagree; they just show they DO NOT (during warming). They do not perform the relevant experiments (pharmacological or during activity) to ascertain whether or not they are physiologically capable of increasing heart rate.

Intrinsic heart rate, on physical principles alone, increases with warming (faster ion diffusion). If in vivo heart rate does not increase, something must be happening to counteract it. Whether this 'something' is 'pathological', so to speak, or actively regulated (e.g. increased vagal tone) is at the crux of the issue, but the authors do not investigate it, and relegate the possible reasons to future work. Unfortunately it is critical for understanding the present work. It is particularly disappointing that the authors say they set out with 'the goal of providing mechanistic insights' (line 61) but in this regard fail to do so.

As the authors are aware, hypoxic bradycardia is well described in fishes and it can normally be mainly attributed to increase vagal tone and therefore appears regulated. It is essential that experiments are performed in vagotomised/atropinised fish. I would suspect this could abolish or attenuate the bradycardia, and then the effects of hypoxia on thermal tolerance would become clearer (i.e. I would predict CT_{max} may be reduced during hypoxia even when bradycardia is prevented- if not, then indeed you might be in a position to conclude the decreased heart rate is mechanistically contributing to limit thermal tolerance).

Another interesting approach could be to electrically pace the hearts of hypoxic fish to attain heart rates similar to normoxic fish at equivalent temperatures- I again would predict this would not rescue CT_{max} during hypoxia. Such experiments would provide 'mechanistic insight'- the present report is purely descriptive.

At present there is nothing more than correlation and I am strongly inclined to believe that HR and CT_{max} are both being diminished by other factors- the reduced tachycardia is not necessarily responsible for the reduced CT_{max}

I think it is problematic the way the authors generalise from the current investigation on a single species to an all-encompassing title. It should better reflect that this was done on a single species.

It is not sufficient to, at the end of the manuscript, cite unpublished data in other species to make this claim. If the authors wish to make more broad generalisations, I suggest they combine the data sets to make a more compelling case for generality.

Line 187: "The scope for $\dot{M}O_2$ was calculated as maximum - routine $\dot{M}O_2$. We are aware that this "temperature-induced" aerobic scope (AST) is not the same metric as that determined with traditional swimming-flume and chase protocols"

It is still misleading and completely confusing to refer to $\dot{M}O_2$ this way as 'maximum' which implies some form of physical activity.

Surely these fish do not truly have a negative aerobic or cardiac scope in its common sense. Much of the terminology in fish respirometry is complicated enough, the last thing we need is new definitions being ascribed to 'maximum $\dot{M}O_2$ ' or aerobic scope.

Minor comments:

Line 42: "cardiac collapse largely sets the fishes' upper thermal limits". This is unbalanced, the evidence is highly controversial and other mechanisms, e.g. neuronal failure, are plausible (e.g.

Jutfelt et al. 2019, doi: 10.1242/jeb.208249). The heart seems to ‘largely’ set the upper limit if you only study the heart.

Line 276: “must have been associated with the splenic release of RBCs”. It could also be due to haemoconcentration (e.g. Hedrick et al., 2020, doi: 10.1242/jeb.223586)

Line 276 again: ‘However, it is clear that the enhancement’. It is not clear unless you measure CaO₂ and CvO₂.

Line 306: “All of these are discussed below.” I recommend deleting this sentence.

Supplementary methods: “none of the probes displayed significant temperature sensitivity.” Temperature sensitivity is an inherent property of these probes, I am surprised to see how they cannot be temperature sensitive over the broad temperature range in the present study (as stated by the manufacturer, ‘Acoustical velocity increases with temperature increases’, <https://www.transonic.com/index.cfm/tasks/render/file/?fileID=E30B0E3D-C2A1-7D3C-02AF89D27453791A>)

Review form: Reviewer 5

Recommendation

Accept with minor revision (please list in comments)

Scientific importance: Is the manuscript an original and important contribution to its field?

Excellent

General interest: Is the paper of sufficient general interest?

Excellent

Quality of the paper: Is the overall quality of the paper suitable?

Excellent

Is the length of the paper justified?

Yes

Should the paper be seen by a specialist statistical reviewer?

No

Do you have any concerns about statistical analyses in this paper? If so, please specify them explicitly in your report.

No

It is a condition of publication that authors make their supporting data, code and materials available - either as supplementary material or hosted in an external repository. Please rate, if applicable, the supporting data on the following criteria.

Is it accessible?

Yes

Is it clear?

No

Is it adequate?

Yes

Do you have any ethical concerns with this paper?

No

Comments to the Author

RSPB-2020-2340.R1

This paper from K Gamperl's lab mechanistically addresses an important question about the tolerance capacities of fish with the dual stressors of warm temperatures and low oxygen. They combine in vivo cardiorespiratory physiology with blood metrics and whole animal MO₂ recordings with long term acclimation. They have created an impressive data set with which to interrogate their physiologically and environmentally relevant questions. The finding that acute hypoxic strongly reduces the capacity of the CVS to defend against a thermal challenge is novel, important and has wide ranging implications. My comments/suggestions are minor.

1) I think the use of 'fish' in the title is too broad. You cannot generalise 40000 species with an experiment on a single species. The title should specify the species of fish studied. Especially in light of the discussion part (b) where the differences between the well-studied salmonids and the sable/eel like fished are discussed in relation to O₂ extraction and MO₂.

2) Ln 100 it seems odd to restrict the normoxic fish food to the amount eaten by the hypoxic fish. What was the rationale? If both groups were feed ad libitum do you think their morphometrics would be consistent?

3) Ln 109 why is hold temperature and experimental starting temperature different? Is it common to conduct surgery at a much reduced gill irrigation temperature? What is the justification?

4) Figure 2. I find this figure confusing. It took me ages to understand what was being shown despite the fact that the data itself is very clear. I am not sure the schematic of the fish is helpful, not is it explained/mentioned in the legend. The casual reader may think there is only a signal capillary bed of interest in this study. Why is the y-axis legend given as a title not on the left side of each graph? The demarcations on the x-axis make it look like something is missing. I would remove the x-axis marks or put the treatment labels there and put the symbol code in the legend.

5) Figure 4. The figure has demarcations of significance using the GraphPad *, **, *** system but in the legend you only specify a single * is equal to P<0.5.

6) The role of cholinergic tone on the bradycardia is interesting and it is a real shame that pharmacological agents were not included in this study (like that of the Keen paper referenced). It would have provided fascinating insight. Moreover, it was interesting to see an initial increase in V_s with warming during hypoxia, but not in normoxia, and that peak V_s was not overly different regardless of treatment. It would be interesting to unpick the depression of V_s in the initial experiment and the mechanism that allowed it to increase.

7) I like the 'physiological chain of command' argument - points a way to many future studies!

Decision letter (RSPB-2020-2340.R1)

15-Jan-2021

Dear Miss Leeuwis:

Your manuscript has now been peer reviewed and the reviews have been assessed by an Associate Editor. The reviewers' comments (not including confidential comments to the Editor) and the comments from the Associate Editor are included at the end of this email for your reference. As you will see, the reviewers and the Editors have raised some concerns with your manuscript and we would like to invite you to revise your manuscript to address them.

Research ethics:

Use of animals and field studies:

It is a condition of publication that you make available the data and research materials supporting the results in the article (<https://royalsociety.org/journals/authors/author-guidelines/#data>). Datasets should be deposited in an appropriate publicly available repository and details of the associated accession number, link or DOI to the datasets must be included in the Data Accessibility section of the article (<https://royalsociety.org/journals/ethics-policies/data-sharing-mining/>). Reference(s) to datasets should also be included in the reference list of the article with DOIs (where available).

Please submit a copy of your revised paper within three weeks. If we do not hear from you within this time your manuscript will be rejected. If you are unable to meet this deadline please let us know as soon as possible, as we may be able to grant a short extension.

Best wishes,
Dr Daniel Costa
Editor, Proceedings B
mailto:proceedingsb@royalsociety.org

Associate Editor
Comments to Author:

Thanks very much for your patience with the review process, especially over the holidays when we all need a well-deserved break. We have now received two reviews of your revised manuscript. Note that these are not the original reviewers but entirely new reviewers, which we felt was appropriate given the article review history). As you can see, both reviewers raise some critical points that I would like to see you address in a revised version. Thanks again for your patience and flexibility thus far and I look forward to reading a revised version of your manuscript.

Reviewer(s)' Comments to Author:

Referee: 4

Comments to the Author(s)

Leeuwis and colleagues studied the effects of warming on cardiometabolic performance in sablefish during hypoxic versus normoxic conditions. They saw that during hypoxia heart rate and cardiac output did not change during warming, but during normoxia these variables increased. The authors conclude that the inability to increase heart rate and cardiac output during hypoxia constrained thermal tolerance.

Major concerns:

The authors bluntly state, e.g. in the 'running head', that 'hypoxic fish CANNOT increase heart rate'. I disagree; they just show they DO NOT (during warming). They do not perform the relevant experiments (pharmacological or during activity) to ascertain whether or not they are physiologically capable of increasing heart rate.

Intrinsic heart rate, on physical principles alone, increases with warming (faster ion diffusion). If in vivo heart rate does not increase, something must be happening to counteract it. Whether this

‘something’ is ‘pathological’, so to speak, or actively regulated (e.g. increased vagal tone) is at the crux of the issue, but the authors do not investigate it, and relegate the possible reasons to future work. Unfortunately it is critical for understanding the present work. It is particularly disappointing that the authors say they set out with ‘the goal of providing mechanistic insights’ (line 61) but in this regard fail to do so.

As the authors are aware, hypoxic bradycardia is well described in fishes and it can normally be mainly attributed to increase vagal tone and therefore appears regulated. It is essential that experiments are performed in vagotomised/atropinised fish. I would suspect this could abolish or attenuate the bradycardia, and then the effects of hypoxia on thermal tolerance would become clearer (i.e. I would predict CT_{max} may be reduced during hypoxia even when bradycardia is prevented- if not, then indeed you might be in a position to conclude the decreased heart rate is mechanistically contributing to limit thermal tolerance).

Another interesting approach could be to electrically pace the hearts of hypoxic fish to attain heart rates similar to normoxic fish at equivalent temperatures- I again would predict this would not rescue CT_{max} during hypoxia. Such experiments would provide ‘mechanistic insight’- the present report is purely descriptive.

At present there is nothing more than correlation and I am strongly inclined to believe that HR and CT_{max} are both being diminished by other factors- the reduced tachycardia is not necessarily responsible for the reduced CT_{max}

I think it is problematic the way the authors generalise from the current investigation on a single species to an all-encompassing title. It should better reflect that this was done on a single species. It is not sufficient to, at the end of the manuscript, cite unpublished data in other species to make this claim. If the authors wish to make more broad generalisations, I suggest they combine the data sets to make a more compelling case for generality.

Line 187: “The scope for $\dot{M}O_2$ was calculated as maximum – routine $\dot{M}O_2$. We are aware that this “temperature-induced” aerobic scope (AST) is not the same metric as that determined with traditional swimming-flume and chase protocols”

It is still misleading and completely confusing to refer to $\dot{M}O_2$ this way as ‘maximum’ which implies some form of physical activity.

Surely these fish do not truly have a negative aerobic or cardiac scope in its common sense. Much of the terminology in fish respirometry is complicated enough, the last thing we need is new definitions being ascribed to ‘maximum $\dot{M}O_2$ ’ or aerobic scope.

Minor comments:

Line 42: “cardiac collapse largely sets the fishes’ upper thermal limits”. This is unbalanced, the evidence is highly controversial and other mechanisms, e.g. neuronal failure, are plausible (e.g. Jutfelt et al. 2019, doi: 10.1242/jeb.208249). The heart seems to ‘largely’ set the upper limit if you only study the heart.

Line 276: “must have been associated with the splenic release of RBCs”. It could also be due to haemoconcentration (e.g. Hedrick et al., 2020, doi: 10.1242/jeb.223586)

Line 276 again: ‘However, it is clear that the enhancement’. It is not clear unless you measure CaO₂ and CvO₂.

Line 306: “All of these are discussed below.” I recommend deleting this sentence.

Supplementary methods: “none of the probes displayed significant temperature sensitivity.” Temperature sensitivity is an inherent property of these probes, I am surprised to see how they cannot be temperature sensitive over the broad temperature range in the present study (as stated

by the manufacturer, 'Acoustical velocity increases with temperature increases',
<https://www.transonic.com/index.cfm/tasks/render/file/?fileID=E30B0E3D-C2A1-7D3C-02AF89D27453791A>)

Referee: 5

Comments to the Author(s)

RSPB-2020-2340.R1

This paper from K Gamperl's lab mechanistically addresses an important question about the tolerance capacities of fish with the dual stressors of warm temperatures and low oxygen. They combine in vivo cardiorespiratory physiology with blood metrics and whole animal MO₂ recordings with long term acclimation. They have created an impressive data set with which to interrogate their physiologically and environmentally relevant questions. The finding that acute hypoxic strongly reduces the capacity of the CVS to defend against a thermal challenge is novel, important and has wide ranging implications. My comments/suggestions are minor.

- 1) I think the use of 'fish' in the title is too broad. You cannot generalise 40000 species with an experiment on a single species. The title should specify the species of fish studied. Especially in light of the discussion part (b) where the differences between the well-studied salmonids and the sable/eel like fished are discussed in relation to O₂ extraction and MO₂.
- 2) Ln 100 it seems odd to restrict the normoxic fish food to the amount eaten by the hypoxic fish. What was the rationale? If both groups where feed ad libitum do you think their morphometrics would be consistent?
- 3) Ln 109 why is hold temperature and experimental starting temperature different? Is it common to conduct surgery at a much reduced gill irrigation temperature? What is the justification?
- 4) Figure 2. I find this figure confusing. It took me ages to understand what was being shown despite the fact that the data itself is very clear. I am not sure the schematic of the fish is helpful, not is it explained/mentioned in the legend. The casual reader may think there is only a signal capillary bed of interest in this study. Why is the y-axis legend given as a title not on the left side of each graph? The demarcations on the x-axis make it look like something is missing. I would remove the x-axis marks or put the treatment labels there and put the symbol code in the legend.
- 5) Figure 4. The figure has demarcations of significance using the GraphPad *, **, *** system but in the legend you only specify a single * is equal to P<0.5.
- 6) The role of cholinergic tone on the bradycardia is interesting and it is a real shame that pharmacological agents were not included in this study (like that of the Keen paper referenced). It would have provided fascinating insight. Moreover, it was interesting to see an initial increase in V_s with warming during hypoxia, but not in normoxia, and that peak V_s was not overly different regardless of treatment. It would be interesting to unpick the depression of V_s in the initial experiment and the mechanism that allowed it to increase.
- 7) I like the 'physiological chain of command' argument - points a way to many future studies!

Author's Response to Decision Letter for (RSPB-2020-2340.R1)

See Appendix B.

Decision letter (RSPB-2020-2340.R2)

08-Feb-2021

Dear Miss Leeuwis

I am pleased to inform you that your Review manuscript RSPB-2020-2340.R2 entitled "Limited capacity to increase heart function may leave hypoxic fish susceptible to heat waves" has been accepted for publication in Proceedings B.

We do have one final request. The referee(s), the Associate Editor and I all agree that you should put the species of fish in the title. The current title makes the paper seem to be a general treatment of fish in general. Adding the fish name at the end would put the work in a better context.. Once you have made this change, please proof-read your manuscript carefully and upload your final files for publication. Because the schedule for publication is very tight, it is a condition of publication that you submit the revised version of your manuscript within 7 days. If you do not think you will be able to meet this date please let me know immediately.

To upload your manuscript, log into <http://mc.manuscriptcentral.com/prsb> and enter your Author Centre, where you will find your manuscript title listed under "Manuscripts with Decisions." Under "Actions," click on "Create a Revision." Your manuscript number has been appended to denote a revision.

You will be unable to make your revisions on the originally submitted version of the manuscript. Instead, upload a new version through your Author Centre.

1) A text file of the manuscript (doc, txt, rtf or tex), including the references, tables (including captions) and figure captions. Please remove any tracked changes from the text before submission. PDF files are not an accepted format for the "Main Document".

2) A separate electronic file of each figure (tiff, EPS or print-quality PDF preferred). The format should be produced directly from original creation package, or original software format. Please note that PowerPoint files are not accepted.

3) Electronic supplementary material: this should be contained in a separate file from the main text and the file name should contain the author's name and journal name, e.g `authorname_procb_ESM_figures.pdf`

All supplementary materials accompanying an accepted article will be treated as in their final form. They will be published alongside the paper on the journal website and posted on the online figshare repository. Files on figshare will be made available approximately one week before the accompanying article so that the supplementary material can be attributed a unique DOI. Please see: <https://royalsociety.org/journals/authors/author-guidelines/>

4) Data-Sharing and data citation

It is a condition of publication that data supporting your paper are made available. Data should be made available either in the electronic supplementary material or through an appropriate repository. Details of how to access data should be included in your paper. Please see <https://royalsociety.org/journals/ethics-policies/data-sharing-mining/> for more details.

<http://datadryad.org/submit?journalID=RSPB&manu=RSPB-2020-2340.R2> which will take you to your unique entry in the Dryad repository.

Once again, thank you for submitting your manuscript to Proceedings B and I look forward to receiving your final version. If you have any questions at all, please do not hesitate to get in touch.

Sincerely,
Dr Daniel Costa
Editor, Proceedings B
mailto:proceedingsb@royalsociety.org

Decision letter (RSPB-2020-2340.R3)

11-Feb-2021

Dear Miss Leeuwis

I am pleased to inform you that your manuscript entitled "Research on sablefish (*Anoplopoma fimbria*) suggests that limited capacity to increase heart function leaves hypoxic fish susceptible to heat waves" has been accepted for publication in Proceedings B.

Open Access

Paper charges

You are allowed to post any version of your manuscript on a personal website, repository or preprint server. However, the work remains under media embargo and you should not discuss it

with the press until the date of publication. Please visit <https://royalsociety.org/journals/ethics-policies/media-embargo> for more information.

Sincerely,
Editor, Proceedings B
<mailto:proceedingsb@royalsociety.org>

Appendix A

Dear Prof. Costa and other Editorial Board Members,

Enclosed with this document you will find a revised version of the manuscript (RSPB-2020-2340) titled “Limited capacity to increase heart function may leave hypoxic fish susceptible to heat waves” for your consideration as an article for publication in *Proceedings of the Royal Society B: Biological Sciences*. We are grateful for the constructive comments of the three Referees and the Associate Editor on the original manuscript, and we have worked diligently to resolve all issues raised to the best of our ability. We have made the majority of the requested/suggested changes to the manuscript, and in the few cases where an adjustment was not possible/made, we have provided a detailed explanation. Given the revisions that have been made to the manuscript through this process, we now feel that this manuscript has been improved significantly, and that it is now suitable for publication. Below, we respond (**in bold**) to each of the points/queries provided by the Reviewers and Editors, and refer to specific line numbers in the manuscript (with changes tracked and attached to this letter) whenever possible and add full citations for any literature used in our responses.

Responses to Editors

Associate Editor

Comments to Author

Thank you for your patience with the review process. As you can see we have received two expert reviews of your manuscript. Both of these reviewers saw merit in the topic and your approach, and I agree. This is a nicely written manuscript that would be of interest to fish cardiovascular biologists and thermal biologists. Unfortunately, both reviewers provided detailed comments on their concerns with the work. One reviewers in particular raised several critical issues with the manuscript, and one of these major concerns undermines the conclusions in the paper. Specifically, with the data presented in the manuscript, causality is difficult to discern and the reviewer has requested additional data (measurements of oxygen concentrations and partial pressures in the arterial venous blood) to support or refute your interpretations. I agree that this additional information would significantly improve the quality of the work and the reliability of the conclusions drawn. I am well aware that given the ongoing global crisis, additional experiments are a lot to ask, but without these additions the conclusions drawn from the work are not well grounded in the evidence presented, and the work is thus not a good fit for Proc. B.

Authors' response: We thank the Associate Editor for the supportive comments and constructive criticism provided after the response from Reviewers 1 and 2 was received. Because these two reviews were very different in the level of support for the manuscript (i.e., Reviewer 1 expressed a number of concerns, while Reviewer 2 was overall quite positive), we are grateful that the decision was made to send our manuscript out to a third Reviewer. It is important to note that the role of the fish's cardiorespiratory system in thermal tolerance is a very hotly debated subject area, with individual researchers holding a variety of positions/opinions, so we expected that this work would trigger some scepticism. However, this does not detract from the fact that this study is an important

contribution to the field, with novel findings that will be of interest to the *Proceedings B* readership because of their key implications for fish physiology and survival in these times of climate change. With regards to the Associate Editor's comments concerning arterial/venous O₂ measurements, we refer you to our responses to Reviewer 1 in which we have addressed this topic in detail.

Responses to Reviewers

Reviewer 1

Comments to the Author(s)

This study provides interesting data demonstrating that the normal temperature-induced rise in the rate of oxygen consumption, as well as heart rate and cardiac output do not occur when sablefish are exposed to hypoxia. This pattern was similar in fish that had been acclimated to hypoxia for 4-6 months. The study also shows that hypoxia causes a small reduction in the upper critical temperature (CT_{max}) as well as the temperature where heart rate become arrhythmic. The study therefore shows that hypoxia (40% of normal) impairs temperature tolerance, a non-controversial finding with importance for understanding the consequences of current global change scenarios where hypoxia and elevated temperatures often co-occur.

Authors' response: We thank the Reviewer for the helpful comments and constructive criticism. Below, we outlined our responses, point by point, to each of the comments.

Main objections

1. The authors seem to interpret the findings as if a failure of the cardiovascular system constrains aerobic metabolism, measured as oxygen uptake, but the causality is more likely to be the other way around. Given that metabolism of the hypoxic fish does not increase as much as expected with increased temperature, there are virtually no cardiovascular response (consequently MO₂/Q is unaffected by hypoxia). One way to resolve the "hen-egg-problem" of this discussion would have been to measure oxygen concentrations and partial pressures in the arterial venous blood to establish whether hypoxia in fact leads to so low tissue oxygen delivery that metabolism is constrained. This was not done. Instead, the authors estimate arterial venous extraction from the Fick equation (which obviously is perfectly valid if all measurements are correct), but they never relate this calculated extraction to a realistic arterial oxygen concentration that could have been estimated from the Hb concentration and an assumption of oxygen saturation. I'm aware this requires knowledge on the oxygen equilibrium curve of whole blood, but this could have been obtained). The lactate measurement were unfortunately only reported for the highest temperatures and not therefore provide insight to in sufficient oxygen delivery.

Authors' response: We thank the Reviewer for this detailed comment, which consists of four components, which we will respond to in order.

First, we would like to respond to the “hen-egg-problem” that the Reviewer describes. This case of reversed causality, although an interesting concept to explore, is highly unlikely for the following reasons.

(i) Because of the basic thermodynamic effects of temperature on cellular processes and on metabolic rate in ectotherms, exposure of fish to warming increases their O₂ demand. This is true whether fish are under conditions of normoxia or hypoxia. Thus, the sablefish in this experiment that were exposed to hypoxic warming would have had the same metabolic (O₂) demands as the sablefish exposed to normoxic warming. The fact that under hypoxia the sablefish’s O₂ *consumption* did not increase as much with temperature as during normoxia, does not mean that the animal’s O₂ *demands* were lower; there is a difference between consumption and demand. In the case of the hypoxic sablefish, there must have been a mismatch between the O₂ demand vs. supply to the tissues.

(ii) When a fish’s O₂ demands are increased due to warming, the cardiovascular system responds to help supply more O₂ to the tissues, whereby increases in \dot{Q} are key. There are numerous publications that show this, such as Pörtner 2010, Wang *et al.* 2007, Farrell *et al.* 2017, and Eliason *et al.* 2017 (all these references are cited in the introduction of the manuscript). In this study, this is exactly what we observed in sablefish when this species was exposed to normoxic warming (i.e., increase in $\dot{M}O_2$ and \dot{Q}). Only when hypoxia co-occurred with warming, were sablefish unable to enhance \dot{Q} , and this had negative consequences on their ability to raise $\dot{M}O_2$ and their CT_{max}. This indicates/strongly suggests that this is not a normal or adaptive response.

(iii) We have recently performed similar experiments in Atlantic salmon, a more typical teleost fish model than the sablefish, and the results support those obtained in this study (i.e., hypoxia limits the cardiovascular system’s response to warming, which impairs maximum $\dot{M}O_2$ and CT_{max}). We refer to this recent work as a manuscript in preparation in the discussion (Lines 394-395), as this data underlines that our findings in sablefish are generally relevant for teleost fishes.

Based on these arguments, the most logical way to interpret our results is the following order of causality:

Exposure to warming → increases in a fish’s metabolic rate / O₂ demand → due to hypoxia, the cardiovascular system fails to respond: no rise in \dot{Q} → compromised O₂ delivery → reduced $\dot{M}O_2$ and CT_{max}

We will now discuss the Reviewer’s criticism of our use of the re-arranged Fick equation as a method to estimate arterial-venous O₂ extraction, rather than having performed direct measurements. We agree that the latter approach is preferable, given that the indirect method does not account for cutaneous respiration which may result in an overestimation of the arteriovenous difference, and we appreciate that the Reviewer brought this to our attention. We have now modified the text in the Material & Methods where the Fick equation is discussed (Lines 181-184) to acknowledge the method’s limitations, and we included a new reference (Farrell *et al.* 2014, see below) about this specific topic. However, the possible overestimation error should be relatively minor

(given cutaneous O₂ uptake is small compared to the whole-animal O₂ uptake) and would be consistent between acclimation groups and treatments. Therefore, we believe that the Fick calculation still provides a valid and useful estimate.

Furthermore, this indirect method is well-established and commonly used by various world-leading research groups in the field of fish cardiovascular physiology (e.g., see recent work by Clark *et al.* 2006 in *J. Exp. Biol.*; Claësson *et al.* 2016 and Joyce *et al.* 2018 in *Conserv. Physiol.*; Motyka *et al.* 2018 in *J. Therm. Biol.*; and Harter *et al.* 2019 in *Proceedings B*). To clarify this to the reader, we have slightly re-worded Lines 184-185, and inserted some of the aforementioned references that were not already cited. Thus, while we recognize that information on arteriovenous O₂ contents would have been helpful, the claim that our data and conclusions are unreliable is not substantiated.

Finally, measurements of arterial and venous blood O₂ contents would require the fish to be equipped with an additional cannula (in the ventral aorta), and a larger volume of blood to be withdrawn, and these invasive surgical and procedural interventions have impacts on the fish's physiology and performance, and can affect the quality of the results. Reviewer 1 will likely agree that blood sampling should be kept to a minimum, given the Reviewer's concern about the amount of blood taken (see minor comments). We took the additional step to confirm that our current surgical procedures and blood withdrawal did not impact the O₂ consumption and thermal tolerance of the fish (see supplementary material; a step often not done by many authors publishing in this field). However, expanding on/adding to the procedures that were performed, to allow for additional O₂ extraction measurements, can increase the risk of a confounding effect on the primary research outcome(s) (i.e., cardiac function, O₂ consumption, thermal tolerance).

Farrell AP, Eliason EJ, Clark TD, Steinhausen MF. 2014 Oxygen removal from water versus arterial oxygen delivery: calibrating the Fick equation in Pacific salmon. *J. Comp. Physiol. B* 184, 855–864. (doi:10.1007/s00360-014-0839-7)

Clark TD, Seymour RS. 2006 Cardiorespiratory physiology and swimming energetics of a high-energy-demand teleost, the yellowtail kingfish (*Seriola lalandi*). *J. Exp. Biol.* 209, 3940–3951. (doi:10.1242/jeb.02440)

Joyce W, Axelsson M, Egginton S, Farrell AP, Crockett EL, O'Brien KM. 2018 The effects of thermal acclimation on cardio-respiratory performance in an Antarctic fish (*Notothenia coriiceps*). *Conserv. Physiol.* 6, coy069. (doi:10.1093/conphys/coy069)

The Reviewer then points out that a realistic arterial oxygen concentration could have been estimated from the Hb concentration and an assumption of oxygen saturation, and to determine the latter, the Reviewer suggests that the oxygen equilibrium curve of whole blood could have been obtained. However, estimating a “realistic” arterial oxygen concentration for the sablefish in the experiment is actually very difficult to do, given it relies on an assumption of Hb-O₂ saturation, which is heavily influenced by temperature, pH and CO₂ levels (through the Root and Bohr effects, respectively). These parameters

likely changed considerably, and in a complex manner in the arterial blood throughout the experiment given the multi-stressor design, whereby the specific pH and CO₂ conditions *in vivo* were unfortunately not recorded. Furthermore, while Rummer *et al.* 2010 reports the effect of either pH or CO₂ on the Hb-O₂ saturation of whole blood in sablefish (reference already included in the manuscript), the effect of temperature is not known, and the interactions between the O₂ level, temperature, pH, and CO₂ concentration, will also need to be characterised (which requires *in vitro* experimentation). In other words, we currently do not have enough information about how the experimental conditions were affecting the O₂ equilibrium curves of whole blood in sablefish to make a meaningful estimate of the arterial O₂ concentration. We have included a sentence in the manuscript to explain this to the reader (Lines 187-189). Nevertheless, we appreciate the Reviewer's comment and we are very interested to explore this topic in our future research.

Finally, the Reviewer mentioned that the lactate measurements were only reported for the highest temperatures, and therefore do not provide insight in sufficient oxygen delivery. However, we did also report lactate levels at lower temperatures, including 12°C under normoxia and hypoxia, and at 18°C under hypoxia; the latter only 6°C above the acclimation temperature and at ~4°C below the CT_{max} (~22°C). So, overall, this is an intermediate temperature during the thermal challenge, and the lactate level at this temperature provides considerable insight into the sufficiency of O₂ delivery. Specifically, the lactate level at 18°C is significantly increased compared to 12°C, which indicates that the sablefish were already relying on anaerobiosis at this point in the experiment. We discuss this in the results, but now that the Reviewer has highlighted it, we have slightly expanded on it (see Lines 282-284). The usefulness of measuring lactate at other low temperatures (i.e., between 12 and 18°C) is somewhat questionable, and would require more blood to be sampled, which may come with the risk of having confounding effects on the fish's physiology (discussed above).

2. There is no mechanistic explanation for the lack of tachycardia when the fish were heated (see minor comments line 284-287), and it is a great shame that atropine was not injected in some animals to address whether it reflects a hypoxic bradycardia, and hence a regulated response.

Authors' response: We do provide several suggestions for possible mechanistic explanations for the lack of tachycardia in the discussion's section titled "possible reasons for the inability to increase f_H when hypoxic". However, we have indeed not identified the underlying mechanism(s) in this study. We fully agree that the nature of the observed phenomenon should be further investigated, which we now mention in Line 313. In fact, we have follow-up experiments planned that are designed to gain more insights into this, and these will include an atropine-injected treatment group. These experiments will be supported by a Company of Biologists Travelling Fellowship and a Canadian Society of Zoologists Research Travel Grant to Ms. Leeuwis (both largely awarded based on the proposed research and its perceived importance), and performed in the Brauner lab at

the University of British Columbia. However, the fact that these experiments are not part of this manuscript is not a reason to object to the publication of the study, and it would be unreasonable to ask for their inclusion before this manuscript can be published. The amount and quality of data already presented (including that in the supplementary file) is clearly deserved of publication in a top tier journal, and the datasets resulting from these additional studies would constitute at least another paper. The present manuscript reports a highly novel phenomenon, and while it leaves a few questions open, this does not detract from the critical importance and novelty of this initial study. The questions that arise from this work will stimulate future research, which is why this manuscript is an important contribution to this field, and the reason for our decision to submit to the flagship journal *Proceedings B*.

3. The title is catchy, but does not adequately address the study. Did you make any perturbation (exercise, stress, drug infusion etc) to verify that cardiac output and heart rate could not be elevated? If not, there is no demonstration of limited capacity; only a demonstration that cardiovascular performance match metabolism.

Authors' response: We would first like to respond to the Reviewer's comment on the title. In this study, we demonstrate in one species of fish (sablefish) that fish under hypoxia have a limited capacity to increase heart function during warming, which leaves them susceptible to heat waves. The title accurately summarizes this finding for the reader. Furthermore, the title is appropriate, given that since the completion of this study, we have conducted similar experiments in Atlantic salmon (a more typical teleost fish model), and the outcomes robustly support those obtained in this study (i.e., once bradycardic, hypoxic salmon are unable to raise f_H when warmed). We refer to this recent work as a manuscript in preparation in our manuscript (Lines 394-395), as this data makes it clear that our findings in sablefish have relevance for teleost fishes in general. Nevertheless, we appreciate the Reviewer's concern about the title, and we understand that more careful wording would be beneficial. Therefore, we have changed the title slightly to express a level of uncertainty in our conclusion, while we wait for results to be obtained from more fish species.

We will now provide a response to the Reviewer's suggestion that we explore whether sablefish can elevate \dot{Q} and f_H during exercise, chasing/handling stress, or after certain drug infusions. We agree with the Reviewer that this would be interesting to investigate, however, it is important to realize that warming is the most relevant perturbation in the context of climate change. Our study was intended to have relevance for the global spread of hypoxia and the increased occurrence of heat waves; therefore, it is justified that we focused on testing sablefish under conditions of hypoxic warming, and that we did not perform experiments involving other perturbations. Please refer to our response to the Reviewer's main objection 2 (see above) for another discussion of the topic of performing additional experimental perturbations.

Finally, regarding the Reviewer's statement that our study only demonstrates that cardiovascular performance matches metabolism, we would like to refer to our response

to the Reviewer's main objection 1 (see above), in which we provide argumentation for how this interpretation of our results is unlikely.

4. The authors propose that lack of a beta-adrenergic response on the red cells may explain why sablefish seemingly do not offload oxygen. This is unlikely as all other vertebrates – none of which are bestowed with adrenergic stimulation of the Na/H exchanger on their red cells - are quite capable of delivering oxygen in hypoxia. Also, your data on MCHC and red cell protein concentration demonstrates clear and significant signs of swelling indicating a fully functional adrenergic stimulation of the Na/H exchanger and the associated water entry.

Authors' response: It appears that the Reviewer misread/misunderstood our discussion of the absence of the β -NHE in sablefish in the manuscript. We do not state that sablefish do *not* offload oxygen, instead, we describe how there is a “*large* enhancement in O₂ extraction in the sablefish” (Lines 352-353). We also mention in the lines beforehand that the increase in $\dot{M}O_2/\dot{Q}$ is only 16% in rainbow trout (Line 339), while approx. 2-fold in sablefish (Line 343), so sablefish clearly have a relatively high capacity to offload oxygen. We then propose that the lack of the β -NHE in sablefish may actually be a mechanism that allows for a large enhancement in O₂ extraction, given that in the absence of β -NHE, a larger reduction in the intracellular pH of red blood cells would be expected to occur, which would drive more O₂ offloading through the Root effect. We have slightly modified the sentence describing this concept (Lines 353-357), to help avoid a similar misunderstanding to arise among other readers.

With regards to our data on MCHC and red blood cell protein concentration, we agree with the Reviewer that these results demonstrate red blood cell swelling, and we do acknowledge this in the results section of the manuscript (Lines 276-277). However, caution is needed when drawing the conclusion that this indicates a fully functional β -NHE response in the sablefish and that this mechanism is responsible for the cellular water entry. We completely understand the Reviewer's line of thought, but the red cell swelling is not *necessarily* evidence of adrenergic stimulation of Na⁺/H⁺ exchange. Reduced arterial O₂ tension and pH can also induce an increase in red cell volume due to the passive movement of water into the cell following an osmotic gradient, and this is likely exacerbated by the rise in temperature which increases plasma membrane fluidity and permeability (e.g., see Nikinmaa 1992). To prove/disprove the presence of β -NHE in sablefish red cells, it is more appropriate to perform *in vitro* experiments whereby, for instance, changes in cellular ion concentrations are monitored (Nikinmaa 1992), and a pharmacological agonist/analogue of adrenaline is used (e.g., isoproterenol or cAMP; see Rummer *et al.* 2010, which is cited in our manuscript, and Perry *et al.* 1996). We had not included these considerations in the manuscript because of the stringent space limitations, and given that is not the main focus of the study. Nevertheless, because of this Reviewer's comment, we now briefly address this topic in the manuscript (as other readers may have a similar question) in Lines 357-360 and included Nikinmaa 1992 as a reference.

Nikinmaa M. 1992 How does environmental pollution affect red cell function in fish? *Aquat. Toxicol.* 22, 227–238. (doi:10.1016/0166-445X(92)90042-L)

SF Perry, Reid SG, Salama A. 1996 The effects of repeated physical stress on the β -adrenergic response of the rainbow trout red blood cell. *J. Exp. Biol.* 199, 549–562. (doi:

5. The level of hypoxia (40%) needs to be clarified. How does this relate to P_{crit} at the various temperatures. As the authors are aware, P_{crit} increases with temperature and what may be a fully sufficient oxygen level at 12C (i.e. above P_{crit} at that temperature), may be below P_{crit} at 18 and 20C? Again, knowledge on blood oxygen affinity would have been very helpful.

Authors' response: We thank the Reviewer for bringing the level of hypoxia (40% air saturation), and the relationship between P_{crit} and temperature, to our attention. Regarding the hypoxia level, in the introduction (Lines 70-72) we do clarify that 40% air saturation is a moderate hypoxia level for the sablefish, as it is well above the P_{crit} of sablefish (16% air saturation) at its acclimation temperature of 10-12°C. We chose this particular level of hypoxia for the acclimation, because the water O₂ level needed to be low enough to constrain O₂ delivery so that it may trigger a plastic response, but also mild enough that normal behaviour and feeding of the fish was maintained. We have modified the sentence in Lines 70-73 to include some of this information, as to further clarify the level of hypoxia used in the experiment.

Regarding the relationship between P_{crit} and temperature, we are indeed aware that P_{crit} increases with temperature in fish, which is for instance demonstrated by Remen *et al.* 2013 in Atlantic salmon (reference cited below) and Ern *et al.* 2016 in lumpfish and red drum (reference already cited in manuscript). Therefore, it is quite possible that 40% air saturation, while sufficient for sablefish to maintain routine metabolic rate at 12°C, is below the P_{crit} at higher temperatures such as 18 and 20°C; however, we currently do not have experimental data for sablefish to support this. Characterising how the P_{crit} relates to temperature is outside the scope of this study and a study/publication in itself. Although, we have added a sentence to the introduction (Lines 83-85) to acknowledge this relationship (including a reference to Remen *et al.* 2013) and how it may have affected the sablefish in our experiment.

Remen M, Oppedal F, Imsland AK, Olsen RE, Torgersen T. 2013 Hypoxia tolerance thresholds for post-smolt Atlantic salmon: Dependency of temperature and hypoxia acclimation. *Aquaculture* 416–417, 41–47. (doi:10.1016/j.aquaculture.2013.08.024)

6. The discussion give the impression that a change in dogma is required to understand the influence of temperature on cardiorespiratory physiology and metabolism. I completely disagree. A rise in arterial venous extraction with increased temperature was established by Heath and Hughes (JEB 1973), and has been shown in other species (some of the cited in the submitted ms). Also, given that the submitted ms did NOT measure arterial or venous oxygen, the claim for a new dogma seems excessive to me.

Authors' response: It appears that the Reviewer has somewhat misinterpreted this part of our discussion. We did not intend to claim that a rise in arterial-venous extraction with increasing temperature was a dogma-challenging finding. With the “dogma”, we meant to refer to the current and common view in the field of fish cardiorespiratory physiology that increases in \dot{Q} are the most important driver of enhanced O₂ delivery during warming, and the main determinant of upper thermal tolerance. Examples of leading publications that subscribe this view are the review by Farrell *et al.* 2009 in *Can. J. Zool.* (we have added this new reference to the manuscript), and Wang *et al.* 2007 and Eliason *et al.* 2011 in *Science*. The new perspective that we present in the discussion, is that this is not always true: in some species, such as the sablefish, enhanced O₂ extraction is just as important as increases in \dot{Q} in improving O₂ delivery during warming. In other words, our findings suggest that the relative importance of cardiac function vs. arterial-venous extraction in determining thermal tolerance in fish needs to be re-assessed.

However, we realize that we do need to clarify, and soften, our statements in this part of the discussion, to avoid the impression of making an excessive claim / overstating our results, and to ensure that we correctly communicate our message to the reader. We tried to make the manuscript as interesting as possible to the broader audience of *Proceedings B*, and this is why we used wording like “challenging the dogma”, but we have now adjusted this section to take a more careful approach (Lines 331-346).

Farrell AP, Eliason EJ, Sandblom E, Clark TD. 2009 Fish cardiorespiratory physiology in an era of climate change. *Can. J. Zool.* 87, 835–851. (doi:10.1139/Z09-092)

Minor comments (line numbers)

48. A reference to support this claim is needed, also heart function is rather ambiguous (contractility, rate or?).

Authors' response: We are unable to provide a reference for this sentence, because it describes one of the main knowledge gaps prior to this study: that is, the effect of hypoxia on the fish's cardiac response to warming. The sentence was more intended as a hypothesis, not a claim. However, we have added the wording “theoretically” to the sentence to ensure that this is clear to the reader (Line 47-48). We have also replaced “heart function” by “cardiac function (f_H and \dot{Q})” to avoid ambiguity and to provide a more specific description.

54. “vital” is rather excessive, consider to tone down the importance of your own contribution?

Authors' response: We have replaced “vital” with “important” (Line 57), which should be more modest wording, while still indicating the relevance of the study to the reader.

55-56. something is missing in this sentence? What does “its” refer to?

Authors' response: In this sentence, “its” refers to the first part of the sentence, “how physiological plasticity is related to a fish's tolerance to environmental change”. To

clarify this/improve the flow of the sentence, we have replaced “and its” by “as this has” (Line 58).

63. please develop the argument why this species is an ideal model.

Authors’ response: We developed the argument for why sablefish is an ideal model for this study, as suggested by the Reviewer (Lines 66-67).

66-69. P_{crit} depends on temperature so please clarify.

Authors’ response: We have already provided clarification about the temperature dependency of P_{crit} in this part of the manuscript (Lines 71 and 83-85) in response to one of the previous comments by the Reviewer (main objection 5).

76. two self-citations to a rearranged of the Fick equation are two too many.

Authors’ response: We agree with the Reviewer that unnecessary self-citations should be avoided, and it was certainly not our intention to purposefully insert a self-citation; rather, we had only included these two references as examples of recent studies that have also used a rearranged Fick equation to estimate O₂ extraction. However, in the Material and Methods section “Measurements of cardiorespiratory function”, we discuss the Fick equation in more detail (in Lines 184-185), and we realize now that this is most appropriate place in the manuscript to cite authors that have previously used this approach. Therefore, we have removed the two references from this sentence in the introduction (Line 82), as requested by the Reviewer.

80 – methods. The size of the fish at the time of measurements should be included

Authors’ response: In the Material & Methods, we refer the reader to table S4 for detailed information about the weight and length of the fish at the time of measurements. However, upon the Reviewer’s suggestion, we have included the average weight of the fish in this part of the Material & Methods (Line 102), so that this information is more accessible the reader.

88. Po₂ does not have the unit of % (it should be kPa, mmHg, torr etc)

Authors’ response: Percent saturation is commonly used in the fish literature to describe the level of oxygen in water. However, kPa, mmHg and Torr are units traditionally used by cardiorespiratory physiologists, which is likely the reason for the Reviewer’s comment. We do provide the PO₂ in one of these units at the start of the paper (Lines 94-96) to help satisfy the entire readership. But for the remainder of the manuscript, we would like to adhere to % air saturation as the unit of choice.

97-98. Tricaine methanesulphonate is normally called MS222. Why do you use TMS?

Authors' response: We thank the Reviewer for bringing this up. We used TMS as an abbreviation for tricaine methanesulphonate, because this anaesthetic is sold in Canada under that name and is commonly referred to as such in the scientific literature (e.g., in Zanuzzo *et al.* 2015, a reference included in the manuscript). However, as the Reviewer points out, it appears that outside of Canada it is more frequently called MS-222, so we have replaced the abbreviation TMS by MS-222 throughout the text (Lines 108, 111 and 147).

129-134 how many blood samples were taken? The replacement of 'volume with saline is probably not sufficient to maintain blood volume because saline distributes quickly to the entire extracellular space.

Authors' response: We answer the Reviewer's question in this part of the Material & Methods in the sentence "Blood (~0.7 mL) was sampled... at four points during the experiment" (Lines 141-144); i.e., four blood samples were taken. We replaced the sampled blood volume with saline to help maintain blood volume as much as possible. However, we agree with the Reviewer that some of this saline may have been lost to the extracellular space, and we have now slightly modified this sentence (Line 145) to acknowledge that this is an imperfect measure. That said, we would like to direct the Reviewer to the supplement which contains further information about blood sampling, in which we describe how we addressed its potential impact on the fish's physiology (Lines 84-95). We also refer the reader to this supplement section in the Material & Methods (Lines 208-209). In the supplement, we describe that the amount of blood withdrawn from each fish was small relative to the fish's blood volume, and that this is unlikely to cause adverse physiological effects, based on the recent recommendations made by Lawrence *et al.* 2020 (reference included in supplement). Furthermore, we verified that blood sampling did not confound the results for thermal tolerance (figure S3) using sham-cannulated fish from which no blood was withdrawn. So overall, we have thoroughly assessed the impact of the blood withdrawal that was part of our experimental protocols, and this should effectively address the above Reviewer's concern.

Lawrence MJ, Raby GD, Teffer AK, Jeffries KM, Danylchuk AJ, Eliason EJ, Hasler CT, Clark TD, Cooke SJ. 2020 Best practices for non-lethal blood sampling of fish via the caudal vasculature. *J. Fish Biol.*, 14339. (doi:10.1111/jfb.14339)

142. describe the setup and delete the self-citation

Authors' response: We referred to the previous study by Petersen *et al.* 2010 for information about the equipment and software used in the experiment, because we have to adhere to a strict page limit for *Proceedings B*, and by using a reference, we were able to shorten the sentence. This was part of our overall strategy to undertake every effort to keep the Material & Methods as concise as possible. So, we had certainly not included this reference as a means of inserting a self-citation.

However, we understand that describing the setup in the main article could be helpful to the reader (as the information would be more accessible), and we thank the Reviewer for making this suggestion. Therefore, we have added the information to the sentence as a replacement of the reference (Lines 155-157).

151. regulation or disturbance?

Authors' response: We are unsure of what the Reviewer meant with this comment. In this sentence, we describe how arrhythmias at high temperatures may be caused “by issues with ionic regulation at the level of the ventricular myocytes”, whereby replacing “regulation” with “disturbance” does not appear to improve the wording. However, by leaving out “issues with”, we were able to change “regulation” into “disturbance”, and this may be what the Reviewer was looking for (Line 165-166).

157. Do not understand “to secure an R^2 of more than 0.90”; the reduction in PO_2 is linear if the rate of oxygen consumption is stable, duration of closure per se has no effect.

Authors' response: We agree with the Reviewer that, if the O_2 consumption of the fish is stable, that the reduction in PO_2 is stable and that the duration of closure per se has no effect. So, we understand the Reviewer's comment and we will provide an explanation below about what we meant with this statement.

We used AutoResp software from Loligo Systems to determine O_2 consumption rates, and we found that adjustments to the duration of respirometer closure can be important in certain situations to ensure that the R^2 calculated by this system was more than 0.90. At the acclimation temperature, the O_2 consumption of the fish was relatively low, and the period of closure needed to be long enough to have a significant reduction in the PO_2 in order for the AutoResp software to reliably calculate the O_2 consumption rate with a $R^2 > 0.90$. At the highest temperatures, on the other hand, the fish's O_2 consumption rate was relatively high, and we shortened the duration of the closed period as much as possible to minimize the decline in PO_2 inside the respirometers (we describe this in Line 172). At this point during the experiment, the fish sometimes struggled/moved in the respirometer, and this could result in an uneven O_2 consumption rate, which can impact the linearity of the decline in PO_2 , and thus, the R^2 . So, in this case, we needed to slightly lengthen the duration of the closed period to allow for a reliable measurement by AutoResp with a $R^2 > 0.90$.

274. I don't think you measured aerobic scope, so how can your study address this question?

Authors' response: In this study, we defined aerobic scope as the difference between the maximum $\dot{M}O_2$ (the highest $\dot{M}O_2$ measured during acute warming) and the routine $\dot{M}O_2$ (the average $\dot{M}O_2$ measured at the acclimation temperature). We did not determine aerobic scope by measuring the maximum $\dot{M}O_2$ using a swimming or a chasing protocol, and we are aware that these are the most common/traditional methods. We describe this

approach in the Material & Methods (Lines 192-195), where we have now expanded on this topic as well to provide more information to the reader in this section. Nevertheless, recent data suggests that aerobic scope determined by heating an animal from its acclimation temperature (i.e., a CT_{max} test) and by swimming it to a critical speed (i.e., a U_{crit} test) give equivalent values for aerobic scope (see response to Reviewer 2).

Thus, technically, we measured the “temperature-induced aerobic scope” and this is what we referred to in the particular sentence that the Reviewer addressed above. To clarify this to the reader, we have added “temperature-induced” in front of “aerobic scope” in this particular sentence (Line 304) and made similar changes elsewhere in the manuscript where aerobic scope is mentioned (e.g., Line 367).

279. delete “important”

Authors’ response: We have deleted “important” from this sentence and replaced it with “several” (Line 309).

284-287. adenosine is one of many possibilities, so no reason to mention to highlight it. I agree with the authors that vagal tone is a more likely explanation. Why did you not inject atropine or performed cardiac vagotomy?

Authors’ response: We agree with the Reviewer that adenosinergic regulation is one of multiple possible reasons for the depression of f_H , however, it really is worthwhile and appropriate to mention in this section. The section’s heading is “possible reasons for the inability to increase f_H when hypoxic”, and while an increase in vagal tone is the most likely explanation, it is important to provide at least one other potential factor that could have played a role, and adenosine is an important example among hormonal regulators that could have been involved. However, we have slightly reworded this sentence (Line 314-315) to indicate to the reader that adenosine is “one of the potential explanations”. In this way, it is highlighted to a lesser extent.

We would like to refer the Reviewer to our response to the Reviewer’s main comment 2 (see above), in which we discuss the topic of atropine injections and other experimental interventions, such as cardiac vagotomy.

Figure 3A is not needed as all data reappear in Figure 3B, Also, the difference in the x-axis between the two figures is rather confusing.

Authors’ response: We thank the reviewer for bringing this to our attention. The purpose of figure 3a was to show the relationship between $\dot{M}O_2$ and \dot{Q} in the fish exposed to hypoxic warming in more detail by enlarging panel b, as in panel b the data points for these specific groups are more difficult view individually. This is why this data appears in both panels, and why the scaling of the x-axes is different between the two panels. However, we understand that showing the same data twice may be redundant, and that

the different x-axes can be confusing. Further, we have also received critical comments about the usefulness of figure 3a by one of the other Reviewers. Because we need to adhere to a stringent page limit for *Proceedings B*, and given that we have expanded various parts of the manuscript in response to the reviews, resulting into a risk of surpassing that limit, we have decided that it would be best to remove panel a from this figure, as its inclusion (and the space it occupies) can no longer be fully justified. We have revised the figure legend accordingly (Lines 651-665).

Reviewer 2

Comments to the Author(s)

This is an interesting study that reveals a link between the inability of sablefish to increase heart rate or cardiac output during warming in hypoxic waters and reductions in oxygen consumption rate and thermal tolerance. They also demonstrate the long-term hypoxia acclimation does not substantially impact the relationship between measured variables. A separate experiment is conducted to show the under normoxic conditions, sable fish respond to warming in a similar manner to most fish. Overall, the study is generally well designed and the analytical approaches employed are appropriate. I don't have any substantial concerns with the study, but I would like the authors to address a few points in the manuscript.

Authors' response: We thank the Reviewer for the supportive comments and constructive criticism. Below, we outline our responses, point by point, to each of the comments.

Obviously, it would have been preferred if the two studies (hypoxia warming and normoxia warming) were conducted at the same time using fish from the same batch. I understand why the normoxia warming experiment was conducted as an additional experiment, but the authors don't really discuss anywhere in the manuscript the potential issues associated with comparing results from experiments conducted on fish from different sources and done, presumably, at different times. Measurements are made and presented for comparison purposes, but there are differences in various parameters (heart rate) between the groups when held under similar conditions. The authors cannot be certain that the results are directly comparable and that needs to be acknowledged in the manuscript and the implications discussed.

Authors' response: We completely agree with the Reviewer that it would have been preferred if the hypoxic and normoxic warming experiments in this manuscript were conducted with fish from the same population. However, while the Reviewer indicates that the potential associated issues were not really discussed anywhere in the manuscript, the Reviewer may have overlooked the section in the supplement in which we do acknowledge this and discuss the implications (Lines 127-134). This information is in the supplement because of the stringent length restrictions of the journal, so we have chosen to refer the reader to this information in the manuscript instead (Lines 229-230). In the supplement we explain that, despite a difference that was observed in resting heart rate, the resting cardiac output of the two groups was the same (no statistical difference), and given that the latter is the most relevant/important parameter for cardiac performance,

we argue that the groups remain sufficiently comparable. We also propose a reasonable explanation here for the difference in heart rate (i.e., it was likely due to the variation in ventricular mass between the batches of sablefish).

Although we would like to move this discussion from the supplement to the main article to increase the accessibility to the reader in response to the Reviewer's comment, we are unfortunately unable to do so, because the manuscript would then almost certainly exceed the strict upper 10-page limit for *Proceedings B*. The original manuscript barely met the page limits, and we have made various small additions to the text in order to satisfy the Reviewer's requests, but this longer section of text (a paragraph consisting of ~170 words) will be difficult to fit in. However, we have now added a specific acknowledgement to the supplement section about the two populations of sablefish being a limitation of our study (Lines 134-138).

Estimates of $\dot{M}O_{2max}$ and calculations of aerobic scope. Have the authors verified that the highest $\dot{M}O_2$ observed during hypoxia warming is indeed equal to $\dot{M}O_{2max}$. Since this is a relatively new species to be investigated in this regard, it would seem a worthwhile endeavour to demonstrate that warm/hypoxic $\dot{M}O_2$ s cannot be increased further using other means.

Authors' response: We have not verified whether $\dot{M}O_2$ under hypoxia cannot be increased using other means, such as chase protocols and swim-flume respirometry, and we agree with the Reviewer that investigating this would be a worthwhile endeavour given that sablefish is a relatively new study species. However, we have two reasons for leaving this for future research and/or that demonstrate that our approach is valid:

(i) In this study, we were interested in the effect of hypoxia on the fish's thermal tolerance and ability to raise $\dot{M}O_2$. In this context, a thermal challenge (CT_{max} test) is the most relevant/appropriate method to measure maximum $\dot{M}O_2$ and aerobic scope. We have added a statement about this in the manuscript's Material & Methods where the "temperature-induced" method for determining maximum $\dot{M}O_2$ and aerobic scope is explained (Lines 192-195). We also included a reference to Paschke *et al.* 2018 here, which is a study in *Front Physiol.* that demonstrated the relevance of the temperature-induced aerobic scope.

(ii) We are aware that chase protocols and swim-flume respirometry (U_{crit} test) are the most common/traditional methods for eliciting maximum $\dot{M}O_2$, with the U_{crit} test the most reliable in species that are good, sustained swimmers (Norin *et al.* 2016) such as sablefish. However, we believe that the temperature-induced maximum $\dot{M}O_2$ and aerobic scope are very close to the values obtained from swim-flume respirometry. For example, Norin *et al.* 2019 in *PeerJ* showed that the temperature-induced and swimming-induced aerobic scope (AS_T and AT_s , respectively) are the same for Atlantic cod. A comparison between different studies with Atlantic cod also demonstrates that this CT_{max} test provides values that are within 10-20% of those measured in an U_{crit} test:

	maximum $\dot{M}O_2$	aerobic scope
Powell et al. 2016**		
swimming test	183.2	96.5
CT _{max} test	146.2	77.9
Gollock et al. 2006 – CT_{max} test	210.8	128.6
Petersen et al. 2010 – U_{crit} test	234.6	152.1

****Note that the measurements in Powell and Gamperl (2016) were done in fish that had a *Loma morhua* infection, and thus, had a reduced aerobic metabolic capacity.**

Norin T, Clark TD. 2016 Measurement and relevance of maximum metabolic rate in fishes. *J. Fish Biol.* 88, 122–151. (doi:10.1111/jfb.12796)

Powell MD, Gamperl AK. 2016 Effects of *Loma morhua* (Microsporidia) infection on the cardiorespiratory performance of Atlantic cod *Gadus morhua* (L). *J. Fish Dis.*, 39, 189–204. (doi:10.1111/jfd.12352)

Paschke K, Agüero J, Gebauer P, Diaz F, Mascaró M, López-Ripoll E, Re D, Caamal-Monsreal C, Tremblay N, Pörtner HO, Rosas C. 2018 Comparison of aerobic scope for metabolic activity in aquatic ectotherms with temperature related metabolic stimulation: a novel approach for aerobic power budget. *Front. Physiol.* 9, phys.2018.01438. (doi:10.3389/fphys.2018.01438)

Norin T, Canada P, Bailey JA, Gamperl AK. 2019 Thermal biology and swimming performance of Atlantic cod (*Gadus morhua*) and haddock (*Melanogrammus aeglefinus*). *PeerJ* 7, e7784. (doi:10.7717/peerj.7784)

Also, on line 173 –I think the authors need to include at warm temperatures in addition to hypoxia for their estimates of Mo₂max.

Authors’ response: We thank the Reviewer for drawing our attention to this missing detail in the sentence, and we have now included it (Line 191) to clarify that maximum $\dot{M}O_2$ was determined as the highest $\dot{M}O_2$ recorded during conditions of hypoxia along with warming.

62/63 & 66 The units used to describe O₂ for the experiment (% air sat) are different than those used to describe the environmental context of the study (MO_X in mg/L). It would be far more convenient for the reader if O₂ was presented in roughly the same units. Even estimating OMZ in % air sat would be helpful.

Authors’ response: We understand the Reviewer’s request, however, in this case it is difficult to convert the units of mg L⁻¹ used to express the O₂ content in OMZs into % air saturation, which is a unit of O₂ partial pressure. The latter depends on the water’s

temperature, salinity, and atmospheric pressure, and these conditions can greatly vary in OMZs (and are unknown to us), making an estimation of the O₂ level in % air saturation quite unreliable. Furthermore, OMZs are generally defined in the literature by having an O₂ content of <2 mg L⁻¹ (e.g., see Breitburg *et al.* 2018; reference already included in the manuscript). Therefore, leaving the unit as is would be appropriate in this context. Nevertheless, we were able to express the acclimation O₂ level in mg L⁻¹ in addition to % air saturation (Lines 94-96). This way, the O₂ levels in the experiment and in the environmental context of the study can still be conveniently compared by the reader. We thank the Reviewer for making this helpful comment.

68. For completeness and readability, it be useful to briefly indicate why surgery was performed.

Authors' response: We have added the reasons for performing surgery on the fish to this sentence (Lines 73-74), as suggested by the Reviewer.

234. There are differences among the hypoxia/temperature treatments in Figure 1c, so “remained at this level” is not fully accurate.

Authors' response: This is true. As is visible in figure 1c, after cardiac output falls during hypoxia, it remains at this level during warming, *until* it declines even further at the highest temperature (22°C). We have added this detail to the sentence (Lines 257-258) so it is completely accurate, and we thank the Reviewer for bringing our attention to it.

243 to 246. I think Figure 3a does a poor job of showing the point the authors are trying to make, primarily because no stats are given to support the point. Comparisons between Figure 1C and 1G does a much better job.

Authors' response: We thank the Reviewer for the insightful comment. The point that we were trying to make in this part of the manuscript was that the typical/anticipated positive relationship between $\dot{M}O_2$ and \dot{Q} in the fish exposed to hypoxic warming was not present (i.e., \dot{Q} does not increase along with $\dot{M}O_2$). We referred to figure 3a to support this point, because it visualises this exact relationship, and we still think that referring to figure 3 is appropriate here. However, a comparison between figure panels 1c and g indeed also allows the reader to draw that conclusion, so we now refer to these figure panels in the text as well (Line 270). Furthermore, although statistical information (i.e., *P*-values) was provided in the legend of figure 3, we now also give this information in the text (Lines 270-272).

274/275. Figure 2 does not explicitly support the “tight” link suggested here. For example, there is no difference in CTmax between the two hypoxic warming groups despite differences in cardiac function. Broadly, I would agree there is a link, but the suggestion that it is a tight causal link should be softened.

Authors' response: We appreciate that the Reviewer brought this to our attention and we concur with the notion that the link between cardiac function and thermal tolerance is present in a broad sense, but that it is not always apparent. Therefore, as the Reviewer suggested, we have softened this statement in the manuscript (Line 303) to reflect this nuance.

Reviewer 3

Comments to the Author(s)

It was a great pleasure to read the manuscript by Leeuwis and colleagues on the interactions between hypoxia acclimation and high temperature on heart function. Here the authors present an impressive 6-month hypoxia acclimation of sablefish followed by invasive cardiorespiratory measurements in a multistressor study design. This is an impressive feat and rare for ecophysiological studies with global change perspectives. Despite my great enthusiasm for the manuscript, I have a list of queries that may improve the presentation of the manuscript.

Authors' response: We thank the Reviewer for the very supportive comments and constructive criticism. Below, we outline our responses, point by point, to each of the comments.

Major comments

1. The introduction reads well, but it does not narrow down to a set of hypotheses that you test in your experiments. You do present one hypothesis that hypoxia acclimation improves CT_{max} (l 52-54), but this is easy to test and does not require an invasive physiological study. I am sure that you had other specific hypotheses in mind when you designed and executed this sophisticated study, so I urge you to enlighten the reader with your motivation for the study. You actually hint to the fact that there are multiple questions/hypotheses (e.g., line 54), but it is not clear to me what these additional hypotheses are.

Authors' response: We thank the Reviewer for making this helpful suggestion, and we have modified the introduction to clarify / include more information about the hypotheses that we had when we began this study (see Lines 47 and 50-52). We have also added a hypothesis for the additional experiment in the Material & Methods (Line 212; this point in the manuscript is more suitable for providing this prediction than the introduction). Overall, we did not always try to predict the direction of an effect; for instance, we expected to see an effect of hypoxia acclimation on the cardiorespiratory response to warming in sablefish based on prior research in rainbow trout (Motyka *et al.* 2017; reference already included in the manuscript), but we did not make a prediction of the specific nature of this effect given the difference in species and experimental O₂ levels (our study: hypoxia, Motyka *et al.*: normoxia). However, even this relatively “open” hypothesis will help enlighten the reader about our motivation for the study.

2. Scope: You calculate scope, such as aerobic scope, as maximum measured in hypoxia across all temperatures minus the average RMR at 12 degree C in normoxia. You need to justify why this calculation is valid:

2.1 Would it not be more valid to use SMR rather than RMR to calculate scope, which you could extract as a lower percentile of MR over the recovery period in the respirometer?

Authors' response: Standard metabolic rate (SMR) is, indeed, typically used to determine a fish's aerobic scope. However, we would argue that using routine metabolic rate (RMR) is more ecologically relevant than using SMR, as fish in the wild likely exhibit a routine level of $\dot{M}O_2$ the majority of the time, whereas SMR is only reached under experimental conditions that entirely restrict the fish's movements and stimulation. This issue is addressed by Rogers *et al.* 2016, a review that also shows that RMR is used far more frequently by researchers to determine P_{crit} (a measure for hypoxia tolerance) than SMR. Another argument for using RMR to calculate scope instead of SMR, is that the other cardiorespiratory parameters (f_H , \dot{Q} , V_s , $\dot{M}O_2/\dot{Q}$) are calculated using the same equation. So, taking the same approach for $\dot{M}O_2$, is more consistent with how we handled the other variables.

Rogers NJ, Urbina MA, Reardon EE, McKenzie DJ, Wilson RW. 2016 A new analysis of hypoxia tolerance in fishes using a database of critical oxygen level (P_{crit}). *Conserv. Physiol.* 4, cow012. (doi:10.1093/conphys/cow012).

2.2 MMR was determined as the maximum MR over the experimental design. However, MMR changes with temperature, so you cannot calculate scope when "SMR" and MMR were measured at different temperatures.

2.3 You need to justify your specific protocol can be used to determine MMR without the inclusion of a chase protocol or exhaustive swimming.

Authors' response: We agree with the Reviewer that maximum metabolic rate (MMR) and aerobic scope are traditionally measured at the same temperature, using means such as chase protocols and swim-flume respiratory, and that our method based on a thermal challenge is different. To make this clear to the reader, we now refer to "temperature-induced aerobic scope" (AS_T) in the manuscript whenever aerobic scope is mentioned (Lines 193, 261, 296, 298, 304, 367, 376, 380).

Given the Reviewer's concern, we have now also provided a justification for the use of the temperature-induced method to determine MMR in the manuscript's Material & Methods section (Lines 192-195). The following are the reasons in more detail:

(i) For this study, which investigates the temperature-dependent cardiorespiratory responses and thermal tolerance of sablefish, the temperature-induced method is very relevant/appropriate to measure MMR and aerobic scope. For a further discussion of this topic, see Leeuwis *et al.* 2019 (reference included in the manuscript).

(ii) Based on recent studies on Atlantic cod and other species, temperature-induced MMR and aerobic scope are often very close to the values obtained from swim-flume respirometry. See: Gollock *et al.* 2006, Petersen *et al.* 2010, Powell *et al.* 2016,

Paschke *et al.* 2018, Norin *et al.* 2019 (references either included in the manuscript, or provided in this letter in response to previous Reviewer comments).

Minor comments

l 31-32: you only show this in one species, so I do not see how your data can justify this extrapolation across coastal species.

Authors' response: We thank the Reviewer for bringing this concern forward, and we have slightly modified this sentence (Lines 30-31) to make a more careful/conservative statement about how our findings in sablefish may apply to other coastal species.

l. 93-94: Also include that heart and spleen mass did not differ.

Authors' response: We have added to the sentence (Line 103) that the cardiac and splenic masses did not differ between these two acclimation groups.

MO₂/Q provides an absolute measure of tissue oxygen extraction, but to get a feeling for how much of the oxygen-carrying capacity (OCC) is extracted at the tissues, you could consider calculate MO₂/Q/OCC. You hint to this on l. 170-171, but I do not see this in the text.

Authors' response: We thank the Reviewer for this comment. In the manuscript (Lines 186-187), we suggested how changes in blood Hb content (OCC) could have contributed to the observed changes in $\dot{M}O_2/\dot{Q}$ (O₂ extraction). Later in the manuscript (in the results section), we revisit this, as we explain that the >2-fold increase in O₂ extraction was partially mediated by the 25% increase in blood Hb content, but must have been primarily due to augmented O₂ uptake by the tissues.

However, we are hesitant to calculate the OCC from the Hb content in this study, and to use the parameter $\dot{M}O_2/\dot{Q}/OCC$ or $\dot{M}O_2/(\dot{Q}*OCC)$. Quantifying the OCC from the Hb content requires knowledge of the amount of O₂ bound per g Hb, which is influenced by the sablefish's Hb-O₂ affinity (P₅₀) and maximum saturation. The kinetics of Hb-O₂ binding vary with temperature, pH and CO₂ levels (through the Root and Bohr effects, respectively), and these parameters likely changed considerably in the arterial blood throughout the experiment. These parameters were not recorded in our study, and their combined effects on the Hb-O₂ saturation of sablefish blood are still unknown. Therefore, we currently do not have enough information to make a reliable estimate of the OCC, and hence, we are not comfortable with calculating OCC for the expression $\dot{M}O_2/(\dot{Q}*OCC)$.

Following lines 186-187, we have inserted a sentence to summarize the above to the reader:

“Blood O₂ content was not estimated given the combined effects of temperature, pH and CO₂ on Hb-O₂ affinity and maximum saturation of sablefish blood are still unknown.”

Figure 1 and 4. The scaling of the x-axis is indicated by color and text. Remove one of them - I suggest removing the color.

Authors' response: We had used color in addition to text in the scaling of the x-axis, to help with the visualisation of the results, with the aim of making it easier for the reader to quickly interpret our findings. The color had the intended message “the darker the red, the warmer the temperature”. However, we understand that using both color and text for the scaling of the x-axis may be redundant. Thus, as suggested by this Reviewer, we have removed the color from the x-axes of figures 1 and 4, as well as from the similarly coloured x-axes in figures S2 and S4.

l. 197: No, the fish in the additional experiment had longer fork length, lower condition factor, and higher heart masses. This should be noted here. You should also mention in the discussion if you expect these differences to affect the interpretations of your results.

Authors' response: We thank the Reviewer for noticing that a mention of the differences in the morphometrics between the fish in the initial and additional experiment is missing in this sentence, and we have now included this information (Lines 218-219). Furthermore, we agree with the Reviewer that these differences and their potential effect on the results need to be discussed. It is unlikely that the differences in length and condition factor affected our results, but the differences in cardiac morphometry may be important and we discuss these in the supplement (Lines 127-134). Because of the stringent space limitations (i.e., a strict upper 10-page limit) for *Proceedings B*, we are unfortunately unable to transfer this information (a paragraph consisting of ~170 words) to the main article's discussion. Otherwise our manuscript would almost certainly exceed the 10-page limit and this can lead to its rejection. The original manuscript barely met the page limits, and we made various small additions to the text in order to satisfy the Reviewer's requests. Please note that we do refer the reader to the extra information about the additional experiment in the main manuscript (Lines 229-230). We have also expanded the discussion in the supplement about the observed differences (Lines 134-138).

What is the purpose of the fish in fig 2 other than showing where the heart is positioned in a fish? I suggest removing it.

Authors' response: The purpose of the fish in figure 2 is not only to show the position of the heart, but it is also intended to represent and visualise the three main cardiorespiratory processes/variables that are shown in the panels *a-c* below the illustration. The fish's heart is actually connected with lines pointing to the scope for cardiac function results (*a*), while the lines drawn from the capillaries are directed towards the data for the scope for O₂ consumption and extraction (*b-c*). The main aim of the figure is to summarize how the capacities of the fish to increase cardiac output, O₂ consumption and O₂ extraction, are linked to the animal's thermal tolerance (panel *d*, located at the bottom). In our opinion, including an illustration of a sablefish with a

simplified cardiovascular system helps to convey this message. Furthermore, it could be helpful to those readers that are not familiar with this species of fish, to be shown a general morphological outline. Therefore, while we appreciate the Reviewer's suggestion, we have decided not to remove the fish from this figure.

Very minor comment

You report oxygen partial pressures in %, mg l⁻¹, and kPa. Be consistent throughout the text, and consider using mmHg as many respiratory physiologists may read the paper. Also, consider using mM/μM/nM for concentrations of Hb, lactate, MCHC, glucose, NH₃, and cortisol.

Authors' response: We use % air saturation consistently throughout the text as the unit to describe the O₂ level, and this remains our first choice for this manuscript. We do use mg L⁻¹ once to indicate the O₂ content in OMZs in the introduction (Line 66), and we used kPa twice in the Material and Methods to describe the acclimation O₂ level in addition to providing it in % air saturation (Lines 94-96). Unfortunately, it is difficult to convert mg L⁻¹ to a unit for O₂ partial pressure, like mmHg, given that we do not have information about the temperatures, pressures and salinities found in OMZs. Therefore, we feel this is best left as is. However, we have replaced kPa by mmHg, as suggested by the Reviewer. Furthermore, we now also express the O₂ level in mg L⁻¹ here, so that this unit is more consistently used throughout the manuscript, and it allows the reader to easily compare the acclimation O₂ level with that found in OMZs.

We understand/appreciate the Reviewer's comment with regards to the blood/water parameters measured. We report them in the units that are most commonly used in the fish literature. This allows the reader to easily compare our values with those reported for sablefish elsewhere, or for other species. Thus, we would prefer not to change them, but will do so if this Reviewer insists.

Appendix B

Professor Costa and other Editorial Board Members:

Enclosed with this document, you will find a revised version of the manuscript (RSPB-2020-2340.R1) for your consideration as an article for publication in *Proceedings of the Royal Society B: Biological Sciences*. We thank the two new Reviewers and the Associate Editor for their thoughtful and constructive comments on our manuscript, and we have worked diligently to resolve all issues raised to the best of our ability. We have made the majority of the requested/suggested changes, and in the few cases where an adjustment was not possible/made, we have provided a detailed explanation. Given the revisions that have been made through this process, we feel that the manuscript has been improved significantly, and we hope that it is now suitable for publication. Below, we respond (**in bold**) to each of the statements/queries provided by the Associate Editor and Reviewers, and refer to specific line numbers in the revised manuscript (with changes tracked) whenever possible. We have also included full citation details for any literature used in our responses.

Associate Editor

Comments to Author

Thanks very much for your patience with the review process, especially over the holidays when we all need a well-deserved break. We have now received two reviews of your revised manuscript. Note that these are not the original reviewers but entirely new reviewers, which we felt was appropriate given the article review history. As you can see, both reviewers raise some critical points that I would like to see you address in a revised version. Thanks again for your patience and flexibility thus far and I look forward to reading a revised version of your manuscript.

Response: We completely understand the delay in the review process, and we thank the Associate Editor for the opportunity to address the points raised by the Reviewers in a revised manuscript.

Reviewer 4

Comments to the Author(s)

Leeuwis and colleagues studied the effects of warming on cardiometabolic performance in sablefish during hypoxic versus normoxic conditions. They saw that during hypoxia heart rate and cardiac output did not change during warming, but during normoxia these variables increased. The authors conclude that the inability to increase heart rate and cardiac output during hypoxia constrained thermal tolerance.

Response: We thank the Reviewer for the helpful comments and constructive criticism given below. On the following pages, we provided our responses, point by point, to each of the comments.

Major concerns

The authors bluntly state, e.g. in the ‘running head’, that ‘hypoxic fish CANNOT increase heart rate’. I disagree; they just show they DO NOT (during warming). They do not perform the relevant experiments (pharmacological or during activity) to ascertain whether or not they are physiologically capable of increasing heart rate.

Response: We thank the Reviewer for bringing this to our attention. It is difficult to provide a sufficiently nuanced running head given the journal’s strict limit of 40 characters; otherwise we would have specified “during warming” to indicate to the reader under which condition the fish were tested. However, we agree with the Reviewer that, given this study’s design/dataset, we cannot exclude the possibility that sablefish are physiologically capable of increasing heart rate while hypoxic under a different experimental protocol, such as undergoing pharmacological manipulation or physical activity. Therefore, we have changed the running head to “Hypoxic fish do not increase heart rate”, and we hope that the Reviewer is more comfortable with this version of the running head. Furthermore, in line with the above, we have edited line 26 of the abstract, and made a few adjustments to section (a) of the discussion where the lack of tachycardia was discussed in a similar context:

Lines 314-315 (heading): “Possible reasons why f_H did not increase during hypoxic warming”

Lines 316-317: “A key question, which future studies will need to explore, is why hypoxic sablefish were unable to/did not increase f_H when exposed to rising temperatures (figure 1a, table S3).”

We hope these changes meet the Reviewer’s approval.

Intrinsic heart rate, on physical principles alone, increases with warming (faster ion diffusion). If in vivo heart rate does not increase, something must be happening to counteract it. Whether this ‘something’ is ‘pathological’, so to speak, or actively regulated (e.g. increased vagal tone) is at the crux of the issue, but the authors do not investigate it, and relegate the possible reasons to future work. Unfortunately it is critical for understanding the present work. It is particularly disappointing that the authors say they set out with ‘the goal of providing mechanistic insights’ (line 61) but in this regard fail to do so.

Response: We had used the wording ‘with the goal of providing mechanistic/physiological insights’ in line 61 in a specific context; in the remainder of this sentence, we explain that it was our aim to obtain insights ‘into the potential consequences of hypoxia on the cardiorespiratory response and susceptibility of fish to acute warming events’. With this goal in mind, we measured various parameters of cardiorespiratory function (f_H , Q , V_S , MO_2 , MO_2/Q , and several hematological parameters) during acute hypoxic and normoxic warming, and after hypoxic and normoxic acclimation, and explored how these parameters relate to thermal tolerance. So in this respect, we do provide mechanistic and physiological insights.

However, the Reviewer is correct that we did not elucidate the possible reasons for the lack of an increase of *in vivo* heart rate in sablefish during hypoxic warming. We have now removed the words “mechanistic/physiological” from this sentence (line 64), which hopefully resolves the Reviewer’s concern.

Furthermore, we would like to clarify for the Reviewer that identifying the underlying drivers/factors responsible for the lack of temperature-induced tachycardia was not the purpose of the study. Frankly, we had not anticipated that there would be no/little increase in heart rate or cardiac output in sablefish that were warmed while hypoxic. We agree with the Reviewer that determining the mechanisms involved is key (i.e., whether it is ‘pathological’ or heart rate is actively downregulated), and these experiments are indeed planned (please see below). That said, we strongly believe that the submitted manuscript is more than worthy of being published in *Proceedings B* without this data. Our manuscript reports a highly novel phenomenon, and while it leaves a few questions open, this does not detract from the critical importance and originality of this initial study. The dataset on cardiorespiratory function presented in this comprehensive work, involves fish exposed to both acute and chronic hypoxia, and provides key insights into the capacity of fish to deal with the combined effects of hypoxia and acute increases in temperature. There are no comparable data in the literature.

As the authors are aware, hypoxic bradycardia is well described in fishes and it can normally be mainly attributed to increase vagal tone and therefore appears regulated. It is essential that experiments are performed in vagotomised/atropinised fish. I would suspect this could abolish or attenuate the bradycardia, and then the effects of hypoxia on thermal tolerance would become clearer (i.e. I would predict CTmax may be reduced during hypoxia even when bradycardia is prevented- if not, then indeed you might be in a position to conclude the decreased heart rate is mechanistically contributing to limit thermal tolerance).

Another interesting approach could be to electrically pace the hearts of hypoxic fish to attain heart rates similar to normoxic fish at equivalent temperatures- I again would predict this would not

rescue CT_{max} during hypoxia. Such experiments would provide ‘mechanistic insight’ - the present report is purely descriptive.

Response: We thank the Reviewer for providing examples of additional experiments that could provide mechanistic insights into the cardiac response of sablefish to hypoxic warming. We agree fully with the Reviewer that experiments involving vagotomised/atropinised fish are important as they would allow us to specifically assess the role of cholinergic control in preventing tachycardia during hypoxic warming, and that electrically pacing the hearts of fish exposed to hypoxic warming to attain higher heart rates is another interesting approach. Similar experiments were planned over a year ago, and will be partially funded by a Company of Biologists Travelling Fellowship and a Canadian Society of Zoologists Research Travel Grant to Ms. Leeuwis (both awarded based on the present research, its perceived importance, and the experiments that were proposed). Unfortunately, to date, Covid-19 related university research and travel restrictions have precluded us from completing them.

At present there is nothing more than correlation and I am strongly inclined to believe that HR and CT_{max} are both being diminished by other factors- the reduced tachycardia is not necessarily responsible for the reduced CT_{max}

Response: We feel that it is appropriate, given the large amount of experimental support in the literature, to link the constrained cardiac function (due to a lack of tachycardia) in hypoxic fish with the decline in thermal tolerance (CT_{max}) in the manuscript. There are numerous publications which show that, when a fish’s O₂ demands are inevitably increased due to warming, the cardiovascular system responds to help supply more O₂ to the tissues; cardiac output increases (typically through a rise in f_H) are key in this regard (e.g., see Pörtner 2010, Wang *et al.* 2007, Eliason *et al.* 2017, and Farrell *et al.* 2017; all these references are cited in the introduction of the manuscript). In this study, this is exactly what we observed in sablefish when this species was exposed to normoxic warming (i.e., increase in $\dot{M}O_2$, f_H and \dot{Q}). Only when hypoxia co-occurred with warming, did the sablefish fail to enhance \dot{Q} , and this was associated with large decreases in their $\dot{M}O_2$ and CT_{max}. These data strongly suggest that the severely limited cardiac function was an important contributor to the reduced maximum $\dot{M}O_2$ and thermal tolerance, i.e., that there was more than a mere correlation between the two observations. Specifically, we propose that a reasonable way to interpret our results is the following order of causality:

Exposure to warming → increases in a fish’s metabolic rate / O₂ demand → due to hypoxia, the cardiovascular system fails to respond appropriately: no rise in \dot{Q} → compromised O₂ delivery → tissue hypoxia and reduced CT_{max}

At the same time, we understand the Reviewer’s concern that at this point, we cannot be absolutely certain that there is a direct causal relationship between the reduced \dot{Q} and CT_{max} . Therefore, we have modified our wording in the manuscript slightly to ensure that we avoid implying direct causality. Although we typically do write about our findings in such a neutral way, we have adjusted it in two places (inserted or altered text is underlined):

Lines 25-28 in the abstract:

“We report that sablefish (*Anoplopoma fimbria*) did not increase heart rate or cardiac output when warmed while hypoxic, and that this response was associated with reductions in maximum O₂ consumption and thermal tolerance (CT_{max}) of 66% and ~3°C, respectively.”

Lines 392-395 in section (d) of the discussion:

“In other words, the depression of f_H by hypoxia appears to dominate over the requirement for increasing f_H during warming, and this severely limits the fish’s cardiac response to the latter stressor, and ultimately, appears to constrain thermal tolerance...”

We hope that these changes to the manuscript text, along with the justification provided above, have appropriately addressed this Reviewer’s comment.

I think it is problematic the way the authors generalise from the current investigation on a single species to an all-encompassing title. It should better reflect that this was done on a single species. It is not sufficient to, at the end of the manuscript, cite unpublished data in other species to make this claim. If the authors wish to make more broad generalisations, I suggest they combine the data sets to make a more compelling case for generality.

Response: We thank the Reviewer for bringing this to our attention. Although we understand the Reviewer’s concern, we would prefer to leave the title as is, and would argue that it is appropriate to do so for the following reasons:

- (i) We have formulated our title very carefully to express a level of uncertainty by using the word “may” (“Limited capacity to increase heart function may leave hypoxic fish susceptible to heat waves”). Thus, although the title refers to fish in general, it does not make a rigid claim, in fact the wording is more nuanced. Expressing uncertainty in a title’s statement in this way, when only experimental data from a single species are being reported, has been done before by other authors in other prestigious journals, such as Rummer *et al.* (2013) in *Science*, where the title is “Root Effect Hemoglobin May Have Evolved to Enhance General Tissue Oxygen Delivery” and the study was only performed on rainbow trout.

- (ii) Because the findings of this study are so novel, and have wide-ranging implications for this field of research, we believe that it is important that the title reflects the novelty and importance of the work/its findings. The current title accomplishes this. The title is concise, engaging, and accurately describes our study's results and its potentially broader implications. Furthermore, the present title will help attract a larger readership and spark interest among the general biological research community, which aligns with the goal of *Proceedings B* as a flagship journal.
- (iii) We specify in the abstract, which one is most likely to read after seeing the title, that the study was done on sablefish, and that the findings in this species could apply to other coastal fishes. Thus, readers will very quickly be provided with this information.

Regarding the unpublished data on Atlantic salmon (a more typical teleost fish model), we refer to this work towards the end of the manuscript because they substantiate that our findings in sablefish have relevance for teleost fishes in general. The experiments in Atlantic salmon were similar in design to the present study on sablefish (i.e., either exposure to normoxic or hypoxic warming) and the outcomes robustly support those obtained in this study (i.e., once bradycardic, hypoxic salmon do not raise f_H when warmed, and this limitation in the cardiovascular system's response to warming when hypoxic is also associated with a reduced temperature-induced $\dot{M}O_2$ and CT_{max}). Further, we would like to avoid combining the sablefish and Atlantic salmon datasets, because (i) they are not entirely comparable (e.g., the Atlantic salmon study did not include hypoxic acclimation or the measurement of hematological parameters, but on the other hand, the salmon were exposed to several other levels of hypoxia), and (ii) both datasets are substantial, and it would be unreasonable if both were needed for the publication of the present manuscript.

Line 187: "The scope for $\dot{M}O_2$ was calculated as maximum – routine $\dot{M}O_2$. We are aware that this "temperature-induced" aerobic scope (AST) is not the same metric as that determined with traditional swimming-flume and chase protocols"

It is still misleading and completely confusing to refer to $\dot{M}O_2$ this way as 'maximum' which implies some form of physical activity.

Surely these fish do not truly have a negative aerobic or cardiac scope in its common sense. Much of the terminology in fish respirometry is complicated enough, the last thing we need is new definitions being ascribed to 'maximum $\dot{M}O_2$ ' or aerobic scope.

Response: We agree with the Reviewer that 'maximum $\dot{M}O_2$ ' is most commonly/traditionally measured with swim-flume respirometry (U_{crit} test) and chase protocols (e.g., see Norin *et al.* 2016) which both involve some form of physical activity. However, we do not agree with the

Reviewer that it necessarily *implies* physical activity, as a thermal challenge (CT_{max} test) has also been used to measure maximum $\dot{M}O_2$ and aerobic scope in fish, and the relevance of this method has been demonstrated in the literature. For instance, see Paschke *et al.* (2018) in *Front. Physiol.*, Leeuwis *et al.* (2019) in *Comp. Biochem. Physiol. A*, and Norin *et al.* (2019) in *PeerJ*. We had cited these references in the manuscript in the sentence that the Reviewer had commented on above (lines 193-197):

“We are aware that aerobic scope is most commonly determined using swimming-flumes and/or chase protocols, but the use of “temperature-induced” aerobic scope (AS_T) is appropriate for this thermal tolerance study and provides equivalent data to these traditional methods (e.g., see [27,37,38]).”

Please note that we slightly modified the wording in this sentence to help make the statement clearer to the reader. Below, we also provide two reasons why the use of the temperature-induced maximum $\dot{M}O_2$ and AS are appropriate metrics for this study:

- (i) In this study, we were interested in the effect of hypoxia on the fish’s thermal tolerance and ability to raise $\dot{M}O_2$ in response to warming. In this context, a thermal challenge (CT_{max} test) is the most relevant/appropriate method to measure maximum $\dot{M}O_2$ and AS.
- (ii) The U_{crit} test is the most utilized physical activity-based method of determining maximum metabolic rate and aerobic scope in species that are good, sustained swimmers (Norin *et al.* 2016), and we believe that the temperature-induced maximum $\dot{M}O_2$ and AS reported in this manuscript would be very close to values obtained from a U_{crit} test. For example, Norin *et al.* (2019) in *PeerJ* showed that the temperature-induced and swimming-induced aerobic scopes (AS_T and AT_S , respectively) are the same for Atlantic cod. A comparison between three different studies with Atlantic cod also demonstrates that the CT_{max} test provides maximum $\dot{M}O_2$ values that are within 10-20% of those measured in a U_{crit} test. See Powell *et al.* (2016), and compare Gollock *et al.* (2006) with Petersen *et al.* (2010) (both cited in this manuscript).

References:

Norin T, Clark TD. 2016 Measurement and relevance of maximum metabolic rate in fishes. *J. Fish Biol.* 88, 122–151. (doi:10.1111/jfb.12796)

Powell MD, Gamperl AK. 2016 Effects of *Loma morhua* (Microsporidia) infection on the cardiorespiratory performance of Atlantic cod *Gadus morhua* (L). *J. Fish Dis.* 39, 189–204. (doi:10.1111/jfd.12352)

Finally, we would like to respond to the other concerns brought forward by this Reviewer:

- (i) In the Methods section (lines 190-197), we provide clear definitions of ‘maximum $\dot{M}O_2$ ’ and ‘temperature-induced aerobic scope (AS_T)’, and everywhere in the manuscript where maximum $\dot{M}O_2$ and aerobic scope are mentioned, it is clearly specified that they have been measured using a thermal challenge/are temperature-induced. For instance, see lines 306-307 at the start of the discussion: “Our results (i) provide additional evidence that cardiac function is linked with temperature-induced aerobic scope and thermal tolerance (figure 2)”. Thus, this should not be confusing or misleading for the reader, and we hope that this resolves the Reviewer’s concern.
- (ii) The Reviewer mentioned that “these fish do not truly have a negative aerobic or cardiac scope in its common sense”. Just to clarify: the fish exposed to hypoxic warming had a *positive* (not a negative) aerobic scope, however, one that was much reduced compared to the fish exposed to normoxic warming. Under hypoxic warming, what we observed was that the sablefish had a *negative* (or effectively zero) scope for cardiac output, which is indeed remarkable and surprising, and illustrates how hypoxia can severely limit the cardiac response to a thermal challenge, and thus, the importance/novelty of the data/this manuscript.

Minor comments

Line 42: “cardiac collapse largely sets the fishes’ upper thermal limits”. This is unbalanced, the evidence is highly controversial and other mechanisms, e.g. neuronal failure, are plausible (e.g. Jutfelt et al. 2019, doi: 10.1242/jeb.208249). The heart seems to ‘largely’ set the upper limit if you only study the heart.

Response: We thank the Reviewer for this helpful comment. We had chosen our wording carefully in this particular sentence and indicated with the word “largely” that cardiac collapse is an important, but not exclusive, contributor to the fishes’ upper thermal limits. Furthermore, due to the stringent space limitations in *Proceedings B*, and the fact that our manuscript had already reached the 10-page upper page limit, we had tried to keep the introduction as concise as possible and to avoid going into detail on this topic. Concerning the Jutfelt *et al.* (2019) paper in *J. Exp. Biol.*, the results of this experiment, while interesting, are far from definitive/convincing. That said, Frederik Jutfelt has recently released a more convincing dataset in this area (see reference #11), and this work is now cited in the manuscript. Based on this data, we have modified the sentence (lines 43-45) to read: “... and cardiac collapse [6–10] and neural impairment [11] appear to be key factors in determining

the upper thermal limit of fishes.” We hope that this addresses/resolves the Reviewer’s concern.

Line 276: “must have been associated with the splenic release of RBCs”. It could also be due to haemoconcentration (e.g. Hedrick et al., 2020, doi: 10.1242/jeb.223586)

Response: We agree with the Reviewer that the increase in blood haemoglobin level in the sablefish could have also been due to haemoconcentration, as is argued by Hedrick *et al.* (2020) in *J. Exp. Biol.* Rather than elaborating on the topic as it is not central to the manuscript, we have edited the text accordingly. It now reads: “This was partially mediated by increases in blood haemoglobin levels (by 25%; figure 4a-b), which occurred despite cell swelling that resulted in a decrease in mean cellular haemoglobin concentration (MCHC; figure 4c) and RBC protein levels (figure S4).” (see lines 278-281); i.e., the mention of the splenic release of RBCs has been removed.

Line 276 again: ‘However, it is clear that the enhancement’. It is not clear unless you measure C_{aO_2} and C_{vO_2} .

Response: We agree with the Reviewer that the direct measurement of C_{aO_2} and C_{vO_2} allows for more clear evidence of an enhancement in O_2 extraction than the observation of a rise in $\dot{M}O_2/\dot{Q}$. In this study, we did not measure arterial and venous O_2 content because it would require the fish to be equipped with an additional cannula (in an efferent branchial artery, and thus the loss of additional functional gill surface area) and a larger volume of blood to be withdrawn, and these surgical and experimental procedures can impact the fish’s physiology and performance, and may have affected the quality of the results. We already took the additional step to confirm that our current surgical procedures and blood withdrawal did not have an effect on the O_2 consumption and thermal tolerance of the fish (see supplementary material), and adding C_{aO_2} and C_{vO_2} measurements to the present experimental protocol could have increased the risk of a confounding effect on the primary research outcomes (i.e., cardiac function, O_2 consumption, thermal tolerance).

Overall, we feel that the use of ‘clear’ in the sentence is appropriate, based on the following reasons:

- (i) Using the re-arranged Fick equation to estimate O_2 extraction is a well-established method, and commonly used by various research groups (e.g., see Clark *et al.* 2006 in *J. Exp. Biol.*; Claësson *et al.* 2016 and Joyce *et al.* 2018 in *Conserv. Physiol.*; Motyka *et al.* 2018 in *J. Therm. Biol.*; and Harter *et al.* 2019 in *Proceedings B*). These references (except for Clark *et al.* 2006) are cited in the Methods section where we discuss the widespread use of the Fick equation (lines 185-186).

- (ii) We are aware that the Fick equation does not account for cutaneous O₂ uptake which may lead to an overestimation of O₂ extraction (which we mention in the lines 183-184), however, the possible overestimation error should be relatively minor (given cutaneous O₂ uptake is small compared to the whole-animal O₂ uptake) and would be consistent between acclimation groups and treatments. Therefore, we argue that the Fick calculation still provides accurate information.
- (iii) $\dot{M}O_2/\dot{Q}$ increased by ~150% in both acclimation groups, while blood haemoglobin levels only increased by ~25% (see lines 276-278). Thus, the latter likely played a minor role in increasing O₂ delivery to the tissues, while the majority of this change would have been facilitated by augmented O₂ uptake by the tissues.

Reference: Clark TD, Seymour RS. 2006 Cardiorespiratory physiology and swimming energetics of a high-energy-demand teleost, the yellowtail kingfish (*Seriola lalandi*). *J. Exp. Biol.* 209, 3940–3951. (doi:10.1242/jeb.02440)

Line 306: “All of these are discussed below.” I recommend deleting this sentence.

Response: We have deleted this sentence as suggested by the Reviewer (see line 312).

Supplementary methods: “none of the probes displayed significant temperature sensitivity.” Temperature sensitivity is an inherent property of these probes, I am surprised to see how they cannot be temperature sensitive over the broad temperature range in the present study (as stated by the manufacturer, ‘Acoustical velocity increases with temperature increases’, <https://www.transonic.com/index.cfm/tasks/render/file/?fileID=E30B0E3D-C2A1-7D3C-02AF89D27453791A>)

Response: The Reviewer is correct that Transonic® flow probes are temperature sensitive, and that this is clearly stated by the manufacturer. This is why in our lab the temperature sensitivity of each probe is always tested prior to the experiments, using different biologically relevant flow rates over the temperature range used in the experiments. The calibration procedure that we describe in the supplementary methods is similar to that used in our previous studies (e.g., see Powell & Gamperl 2016 in *J. Fish Dis.*). Our experience is that the temperature sensitivity (and the nature of the temperature-flow relationship) varies considerably between probes due to the manufacturing process; we have even had probes with a negative relationship with temperature in the past. However, we found that the three individual Transonic® probes used in this particular study did not display any significant pattern of temperature sensitivity, and thus, we feel that our statement in the supplement is justified. We have attached an overview of the calibration results below (Figure 1) in case the Reviewer is interested in the details.

Conclusion

- Probe overestimated flow by ~11% on average for all pump speeds and temperatures
- No clear indication of temperature sensitivity

$y = 1.12x - 1.46$ $R^2 = 1.00$	$y = 1.17x - 1.04$ $R^2 = 1.00$	$y = 1.12x - 1.02$ $R^2 = 1.00$	$y = 1.16x - 0.84$ $R^2 = 1.00$	$y = 1.18x - 0.23$ $R^2 = 1.00$
------------------------------------	------------------------------------	------------------------------------	------------------------------------

Trendline formulas

$y = 1.10x + 1.18$ $R^2 = 1.00$	$y = 1.14x - 0.04$ $R^2 = 1.00$	$y = 1.06x - 0.42$ $R^2 = 1.00$	$y = 1.01x + 1.52$ $R^2 = 1.00$	$y = 1.15x + 0.35$ $R^2 = 1.00$
------------------------------------	------------------------------------	------------------------------------	------------------------------------

- Probe overestimated flow by ~11% on average for all pump speeds and temperatures
- No clear indication of temperature sensitivity

$y = 1.11x - 1.84$ $R^2 = 1.00$	$y = 1.27x - 3.25$ $R^2 = 1.00$	$y = 1.24x - 2.70$ $R^2 = 1.00$	$y = 1.20x - 1.30$ $R^2 = 1.00$	$y = 1.27x - 2.80$ $R^2 = 1.00$
------------------------------------	------------------------------------	------------------------------------	------------------------------------

- Probe overestimated flow by ~13% on average for all pump speeds and temperatures
- No clear indication of temperature sensitivity

Figure 1 (see previous page). Overview of calibration results for the three Transonic® probes used in this study. Values are means±s.e.m. and each measurement was taken twice and in a randomized order. Given the relationship between probe readings and actual flow was very similar for each temperature, there was no indication of significant temperature sensitivity in the probes.

Reviewer 5

Comments to the Author(s)

This paper from K Gamperl's lab mechanistically addresses an important question about the tolerance capacities of fish with the dual stressors of warm temperatures and low oxygen. They combine in vivo cardiorespiratory physiology with blood metrics and whole animal MO₂ recordings with long term acclimation. They have created an impressive data set with which to interrogate their physiologically and environmentally relevant questions. The finding that acute hypoxic strongly reduces the capacity of the CVS to defend against a thermal challenge is novel, important and has wide ranging implications. My comments/suggestions are minor.

Response: We thank this Reviewer for the very supportive commentary, and the helpful suggestions given below. On the following pages, we provided our responses, point by point, to each of the comments.

1) I think the use of 'fish' in the title is too broad. You cannot generalise 40000 species with an experiment on a single species. The title should specify the species of fish studied. Especially in light of the discussion part (b) where the differences between the well-studied salmonids and the sable/eel like fishes are discussed in relation to O₂ extraction and MO₂.

Response: We thank the Reviewer for bringing this to our attention. The other Reviewer had a very similar concern, so we would like to refer the Reviewer to our detailed response to this previous comment on p. 5-6 of this document.

2) Ln 100 it seems odd to restrict the normoxic fish food to the amount eaten by the hypoxic fish. What was the rationale? If both groups where feed ad libitum do you think their morphometrics would be consistent?

Response: The reason for matching the amount of feed provided to the normoxic-acclimated fish to that consumed by the hypoxic-acclimated fish, was to ensure that any possible differences in feed intake (and consequently, any potential disparities in growth and body morphometrics) between the two acclimation groups would not be a confounding factor in this study. Chronic hypoxia has a well-known suppressive effect on appetite in fish (e.g., see

Pichavant *et al.* 2001 in *J. Fish Biol.*; Bernier *et al.* 2012 in *J. Exp. Biol.*; Magnoni *et al.* 2018 in *Sci. Rep.*; and Obirikorang *et al.* 2020 in *Comp. Biochem. Physiol. A*), and if feed was provided *ad libitum*, then the nutritional status of the acclimation groups would likely have become dissimilar at the time of the experiments; especially given that the acclimation period was very long (4-6 months). We would also like to note that, despite restricting feed to some extent, the fish in both acclimation groups still grew during the experiment and had a good body condition (i.e., body weight increased from ~850 g to ~1320 g, and condition factor was >1.0 at the time of sampling), as described in the (supplementary) methods. We have also used the experimental practice of ‘feed matching’ in previous experiments in our lab whereby other species (e.g., Atlantic cod and Atlantic salmon) were acclimated to the same level of hypoxia (40% air saturation) (see Petersen *et al.* 2010 in *J. Exp. Biol.* and Harter *et al.* 2019 in *Proceedings B*).

To make this clear to the reader we have added the following text:

Methods, lines 104-105: “This ensured that there were no nutritional or morphometric differences between the two experimental groups.”

Supplementary methods, lines 31-34: “The feed ration provided to the normoxia-acclimated fish was matched with that consumed by the hypoxia-acclimated fish to avoid any potential disparities in growth and body condition between the two groups, given the suppressive effect of chronic hypoxia on appetite in other fish species (e.g., see [2–3]).”

References:

Pichavant K, Person-Le-Ruyet J, Le Bayon N, Severe A, Le Roux A, Boeuf G. 2001 Comparative effects of long-term hypoxia on growth, feeding and oxygen consumption in juvenile turbot and European sea bass. *J. Fish Biol.* 59, 875–883. (doi:10.1006/jfbi.2001.1702)

Bernier NJ, Gorissen M, Flik G. 2012 Differential effects of chronic hypoxia and feed restriction on the expression of leptin and its receptor, food intake regulation and the endocrine stress response in common carp. *J. Exp. Biol.* 215, 2273–2282. (doi:10.1242/jeb.066183)

Magnoni LJ, Eding E, Leguen I, Prunet P, Geurden I, Ozório ROA, Schrama JW. 2018 Hypoxia, but not an electrolyte-imbalanced diet, reduces feed intake, growth and oxygen consumption in rainbow trout (*Oncorhynchus mykiss*). *Sci. Rep.* 8, 4965. (doi:10.1038/s41598-018-23352-z)

Obirikorang KA, Acheampong JN, Duodu CP, Skov PV. 2020 Growth, metabolism and respiration in Nile tilapia (*Oreochromis niloticus*) exposed to chronic or periodic hypoxia. *Comp. Biochem. Physiol. A* 248, 110768. (doi:10.1016/j.cbpa.2020.110768)

3) Ln 109 why is hold temperature and experimental starting temperature different? Is it common to conduct surgery at a much reduced gill irrigation temperature? What is the justification?

Response: The starting temperature of the experiment was slightly higher due to the time required to complete these experiments. They took approx. 11 hours to complete, and starting at 10°C would have added an additional hour. That said, 2°C is a very small change in temperature, and is unlikely to have influenced the results of this study given that the fish were given at least approx. 24 hours to adjust to this temperature change before measurements began.

In our experience, irrigating the gills during surgery with chilled water (i.e., water at a temperature below the holding temperature) aids with the fish's recovery from the procedure. When the fish is on the surgery table, it is exposed to air at room temperature, which can increase the body temperature of the animal, and irrigation with cool/cold water helps to counteract this. Keeping the fish's body cool is important as it reduces the animal's metabolic rate (and hence, its energy and O₂ demands) while undergoing surgery.

4) Figure 2. I find this figure confusing. It took me ages to understand what was being shown despite the fact that the data itself is very clear. I am not sure the schematic of the fish is helpful, nor is it explained/mentioned in the legend. The casual reader may think there is only a signal capillary bed of interest in this study. Why is the y-axis legend given as a title not on the left side of each graph? The demarcations on the x-axis make it look like something is missing. I would remove the x-axis marks or put the treatment labels there and put the symbol code in the legend.

Response: We thank the Reviewer for identifying issues with the data presentation, and for pointing to individual elements that could be clarified or altered. Accordingly, we have prepared a revised version of figure 2. We also explain the purpose of the fish schematic below.

- (i) We have slightly expanded the figure 2 legend (see lines 647-648) to better explain what the schematic of the fish represents. The illustration is intended to visualize the context/organisational level of the cardiorespiratory parameters/processes that were measured in sablefish, and that likely play an important role in the whole-animal oxygen consumption and thermal tolerance. The fish's heart represents cardiac function, while the capillary bed illustrates the organismal level at which O₂ extraction from the blood is taking place (peripheral gas exchange), and where O₂ is being consumed by the tissues. These elements are connected to the data panels a-c accordingly. Taken together, we believe that this visualization can help to engage non-specialist readers, and help them understand the results.

- (ii) We have moved the panel labels to the y-axis of each graph. Furthermore, only the symbols and units are now shown [e.g., “ MO_2 (mg O₂ h⁻¹ kg⁻¹)”] as the symbols are already explained in the legend. We had originally provided them as a ‘title’ for each of the panels, because it is easier to read in a horizontal orientation, and we thought that the reader may find that helpful. However, because of the Reviewer’s comment, we realize that it may be better/less confusing to display them alongside the y-axis, as this is consistent with the formatting of the other three figures.
- (iii) We have removed the demarcations on the x-axis as suggested by the Reviewer, so that it no longer looks like something is missing there. We prefer this solution over the option of putting the treatment labels at the demarcations, and placing the symbol code in the legend. This is because there is not really enough space for these labels in the present figure layout, and the labels would need to be repeated for each of the four panels which is not very efficient. Furthermore, showing the three treatment labels together in a legend in the lower right corner in figure 2 is consistent with figure 3, which contains the exact same type of legend.

5) Figure 4. The figure has demarcations of significance using the GraphPad *, **, *** system but in the legend you only specify a single * is equal to P<0.5.

Response: We thank the Reviewer for bringing it to our attention that we did not explain the demarcations of significance using one, two or three asterisks in the legend of figure 4. We noticed that this information was missing from the figure 1 legend as well. We have now added the appropriate explanation of the asterisk demarcations to the legends of both figures 1 and 4 (see lines 640-642 and 675-676, respectively).

6) The role of cholinergic tone on the bradycardia is interesting and it is a real shame that pharmacological agents were not included in this study (like that of the Keen paper referenced). It would have provided fascinating insight. Moreover, it was interesting to see an initial increase in Vs with warming during hypoxia, but not in normoxia, and that peak Vs was not overly different regardless of treatment. It would be interesting to unpick the depression of Vs in the initial experiment and the mechanism that allowed it to increase.

Response: We completely agree with the Reviewer that the role of cholinergic tone in regulating the bradycardia observed during hypoxic warming is interesting, and that experiments involving pharmacological agents could provide a fascinating insight. We would like to refer the Reviewer to our response to a similar comment by the other Reviewer on p. 3. In that response, we explain (i) why it is reasonable to leave this research for future work,

and (ii) that we actually have follow-up experiments planned that include pharmacological and other manipulations.

The initial increase in V_s followed by a decline during hypoxic warming, which occurred in both acclimation groups and was not observed during normoxic warming, is indeed intriguing. We agree with the Reviewer that it would be interesting to examine the mechanism(s) that allowed V_s to increase before decreasing. We would like to point out, though, that the physiological relevance of this observation is somewhat unclear. The changes in V_s are relatively small (within the range of $\sim 0.05-0.15 \text{ mL beat}^{-1} \text{ kg}^{-1}$). Perhaps, the initial rise in V_s reflects ‘an attempt’ by the sablefish to compensate for hypoxic bradycardia in order to maintain \dot{Q} , analogous to the response in steelhead trout with zatebradine-induced bradycardia in the study by Keen *et al.* (2012). We mention this as a potential explanation in lines 325-332 of the discussion. However, we really do not completely understand what the drivers of the pattern of changes in V_s were, and agree that this could be an exciting direction for future research. Further, we don’t have space/room, given this journal’s word count restrictions, to adequately discuss this observation.

Now that the Reviewer brought our attention to the V_s response displayed in figure 1, we noticed that in the legend we describe that letters indicate significant differences over the experiment, however, we forgot to provide an explanation for the specific use of lower- and upper-case letters. The same information is missing in the figure 4 legend. Therefore, we have modified/added this information to both figure legends (see lines 636-640 and 671-675). We hope that this further clarifies the letter denotations in these figures.

7) I like the ‘physiological chain of command’ argument – points a way to many future studies!

Response: We thank the Reviewer for providing positive feedback to the idea of a ‘physiological chain of command’, and we share the Reviewer’s enthusiasm that this opens the door to future research.